# Injection site vaccinology of a recombinant vaccinia-based vector reveals diverse innate immune signatures

**Jessamine E. Hazlewood**[1‡], **Troy Dumenil**[1‡], **Thuy T. Le**[1], **Andrii Slonchak**[2], **Stephen H. Kazakoff**[3], **Ann-Marie Patch**[3], **Lesley-Ann Gray**[4], **Paul M. Howley**[5], **Liang Liu**[6], **John D. Hayball**[5,6], **Kexin Yan**[1], **Daniel J. Rawle**[1], **Natalie A. Prow**[1,6], **Andreas Suhrbier**[1,7] *

**1** Inflammation Biology Group, QIMR Berghofer Medical Research Institute, Brisbane, Australia, **2** School of Chemistry and Molecular Biosciences, University of Queensland, St Lucia, Australia, **3** Clinical Genomics, QIMR Berghofer Medical Research Institute, Brisbane, Australia, **4** Australian Genome Research Facility Ltd., Melbourne, Australia, **5** Sementis Ltd., Hackney, Australia, **6** Experimental Therapeutics Laboratory, University of South Australia Cancer Research Institute, Clinical and Health Sciences, University of South Australia, Adelaide, Australia, **7** Australian Infectious Disease Research Centre, Brisbane, Australia

‡ These authors share first authorship on this work.
* Andreas.Suhrbier@qimrberghofer.edu.au

**Data Availability Statement:** Datasets and analyses are available in the Supplementary Tables. All raw sequencing data (fastq files) was submitted

## Abstract

Poxvirus systems have been extensively used as vaccine vectors. Herein a RNA-Seq analysis of intramuscular injection sites provided detailed insights into host innate immune responses, as well as expression of vector and recombinant immunogen genes, after vaccination with a new multiplication defective, vaccinia-based vector, Sementis Copenhagen Vector. Chikungunya and Zika virus immunogen mRNA and protein expression was associated with necrosing skeletal muscle cells surrounded by mixed cellular infiltrates. The multiple adjuvant signatures at 12 hours post-vaccination were dominated by TLR3, 4 and 9, STING, MAVS, PKR and the inflammasome. Th1 cytokine signatures were dominated by IFNγ, TNF and IL1β, and chemokine signatures by CCL5 and CXCL12. Multiple signatures associated with dendritic cell stimulation were evident. By day seven, vaccine transcripts were absent, and cell death, neutrophil, macrophage and inflammation annotations had abated. No compelling arthritis signatures were identified. Such injection site vaccinology approaches should inform refinements in poxvirus-based vector design.

## Author summary

Poxvirus vector systems have been widely developed for vaccine applications. Despite considerable progress, so far only one recombinant poxvirus vectored vaccine has to date been licensed for human use, with ongoing efforts seeking to enhance immunogenicity whilst minimizing reactogenicity. The latter two characteristics are often determined by early post-vaccination events at the injection site. We therefore undertook an injection site vaccinology approach to analyzing gene expression at the vaccination site after intramuscular inoculation with a recombinant, multiplication defective, vaccinia-based

to the Sequence Read Archive (SRA), BioProject accession: PRJNA610695.

**Funding:** JEH was supported by an Australian Government Research Training Program scholarship (https://ppl.app.uq.edu.au/content/uq-and-rtp-research-scholarships-procedures) via the Faculty of Medicine at the University of Queensland (UQ). JEH also received funding from the Global Change Institute/Graduate School at UQ (https://graduate-school.uq.edu.au/current-students/global-change-scholars-program). NAP was awarded an Advance Queensland Research Fellowship by the Queensland Government, Australia (https://advance.qld.gov.au/assets/includes/docs/research-fellowships-guidelines.pdf), with co-funding from Sementis (https://www.sementis.com.au/). AS hold an Investigator grant (APP1173880) from the National Health and Medical Research Council (NHMRC) of Australia (https://www.nhmrc.gov.au/). The project was also funded in part by an NHMRC Project grant APP1141421. We also thank Dr J Aylward (Oncolin) and Prof Ed Westaway (Royal Australian Air Force Association) for their kind philanthropic donations. The funders had no role in study design, data collection and analysis, decision to publish, or preparation of the manuscript.

**Competing interests:** I have read the journal's policy and the authors of this manuscript have the following competing interests: NAP, LL and JDH own Sementis shares. JDH is the current CSO of Sementis. AS was a consultant for Sementis. PMH was the previous CEO/CSO of Sementis. LL and NAP have had, and/or currently have, salary and/or project support from funds provided, whole or in part, via Sementis. Sementis had no role in the design and interpretation of the study, or in preparation of the manuscript.

vaccine. This provided detailed insights into *inter alia* expression of vector-encoded immunoregulatory genes, as well as host innate and adaptive immune responses. We propose that such injection site vaccinology can inform rational vaccine vector design, and we discuss how the information and approach elucidated herein might be used to improve immunogenicity and limit reactogenicity of poxvirus-based vaccine vector systems.

## Introduction

A range of vaccine vector systems based on vaccinia virus (VACV) and other poxviruses have been developed, with several sold as products and many more in development and in human clinical trials [1]. These include Modified Vaccinia Ankara (MVA) [2,3], NYVAC [4], ALVAC [5], fowlpox [6], LC16m8 [7], ACAM2000 [8] and raccoonpox [9]. A large series of recombinant MVA (rMVA) vaccines have been evaluated in non-human primate (NHP) studies [10] and in human clinical trials [1,11], with MVA-BN-Filo recently licensed in Europe as part of a heterologous prime-boost Ebola vaccine [12]. MVA is also licensed as a smallpox vaccine (sold as Imvanex/Imvamune). Recombinant poxvirus vector systems have a number of attractive features for vaccine development including a large payload capacity (at least 25,000 base pairs), potential for cold chain-independent distribution, lack of vaccine DNA integration and induction of both cellular and humoral immunity [1]. Nevertheless, a range of strategies are being sought to improve immunogenicity and reduce reactogenicity [2,13–17]. Both these key characteristics of vaccines are largely dictated by the early behavior of the vaccine at the injection site. However, a comprehensive RNA-Seq approach to characterize the post-inoculation injection site responses has not been undertaken for a recombinant poxvirus-based vaccine.

The Sementis Copenhagen Vector (SCV), derived from the Copenhagen strain of VACV, was recently described [18,19]. SCV can replicate its DNA but is rendered unable to generate viral progeny in vaccine recipients by virtue of a targeted deletion of the *D13L* gene that encodes the essential viral assembly protein, D13. Recombinant SCV vaccines are produced in Chinese Hamster Ovary (CHO) cells modified to express D13 and the host range protein, CP77 [19]. A single construct recombinant SCV vaccine encoding the structural gene cassettes of both chikungunya virus (CHIKV) and Zika virus (ZIKV) (SCV-ZIKA/CHIK) was constructed with each polyprotein immunogen driven by the same synthetic strong early late promoter [20], but from two distinct distant loci from within the SCV genome [18]. A dual ZIKV and CHIKV vaccine was deemed attractive as these virus co-circulate in overlapping geographic regions, and can co-infect both mosquitoes and humans [21–23]. SCV-ZIKA/CHIK was shown to protect against CHIKV and ZIKV in a series of mouse models [18]. In NHPs the vaccine also induced neutralizing antibodies against VACV, CHIKV and ZIKV and provided protection against ZIKV viremia [22].

Systems vaccinology uses mRNA expression profiling to gain a detailed molecular understanding of the behavior of vaccines *in vivo*, thereby informing design and development [24]. The approach has been used to understand and predict immunogenicity [25,26], reactogenicity/safety [27,28] and adjuvant activity [29,30]. Most systems vaccinology studies have analyzed peripheral blood post vaccination, as this is readily accessible in humans. However, herein we described RNA-Seq and bioinformatics analyses of injection sites after intramuscular (i.m.) vaccination. Adult wild-type mice were vaccinated with SCV-ZIKA/CHIK and muscles were harvested at 12 hours post vaccination to characterize early injections site innate responses and identify adjuvant signatures. As vector-induced cytopathic effects (CPE) are only just beginning (at least *in vitro*) at 12 hours post infection [19], this was also deemed a

suitable time to investigate expression *in vivo* of both viral vector genes and expression of the recombinant immunogens. Muscle tissue was similarly analyzed on day 7 post i.m. vaccination to determine the persistence of vaccine transcripts and characterize the evolution of injection site inflammatory responses at a time when vaccine-induced adaptive immune responses are being generated. This is also the time when inflammatory lesions develop after VACV vaccination [31–37], with acute transient injection site reactions also the most common adverse reactions observed in clinical trials of MVA [38]. Finally, feet were harvested on day 7 post vaccination to determine whether SCV-ZIKA/CHIK vaccination might be associated with an arthropathy signature. Several MVA vaccine trials reported transient acute arthralgia as an adverse event [38–40], with arthropathy associated with new CHIKV vaccines remaining a standing concern for regulators [41] after the experience with a live-attenuated CHIKV vaccine [42]. The characterization provided herein of vaccine gene expression and innate host immune responses at the injection site provide both a process and insights that may inform future endeavors to improve immunogenicity whilst limiting reactogenicity of poxvirus-based vaccine vectors.

## Results

### RNA-Seq and differential gene expression

Mice were vaccinated i.m with SCV-ZIKA/CHIK or were mock vaccinated with PBS; feet and quadriceps muscles were then harvested at 12 hours and 7 days post vaccination (S1A Fig). Each of the 3 biological replicates comprised pooled RNA from 4 feet or 4 quadriceps muscles from 4 different mice (S1B Fig). Poly-adenylated mRNA was sequenced using the Illumina HiSeq 2500 Sequencer. Per base sequence quality for >93% bases was above Q30 for all samples. The mean total paired-end reads per group ranged from ≈19 to 24 million, with >91.6% of reads mapping to the mouse genome (S1C Fig). Five groups were analyzed in triplicate (i) quadriceps muscles from mock vaccinated mice (MQ), (ii) quadriceps muscles from mice vaccinated with SCV-ZIKA/CHIK taken 12 hours post vaccination (SCV12hQ), (iii) quadriceps muscles from mice vaccinated with SCV-ZIKA/CHIK and taken 7 days post vaccination (SCVd7Q), (iv) feet taken from mice mock vaccinated i.m. (MF) and (iv) feet from SCV-ZIKA/CHIK vaccinated mice taken 7 days post vaccination (SCVd7F). Reads were mapped to the *Mus musculus* genome (mm10) using STAR aligner, with a similar distribution of read counts observed for all samples (S1D Fig). MDS plots showed close clustering of triplicates and clear segregation between MQ, SCV12hQ and SCVd7Q (S1E Fig). Differentially expressed genes were generated for MQ vs SCV12hQ (for early post-vaccination injection site responses), MQ vs SCVd7Q (for injection site responses on day 7 post-vaccination) and MF vs SCVd7F (to evaluate potential arthritogenic side effects associated with vaccination) (Smear plots are provided in S1F Fig).

### Read alignments to the SCV-ZIKA/CHIK vaccine genome

Expression of the vaccine vector and recombinant ZIKA and CHIK immunogen genes was analysed by aligning reads to a combined reference that included mouse, VACV, ZIKV and CHIKV genomes. Given vaccine transcripts can only be expressed in host cells, SCV-ZIKA/CHIK vaccine reads were expressed as a percentage of total RNA sequencing reads mapping to the mouse genome (Fig 1A and S1A Table). The overall expression profile remained similar when an alternative aligner was used (S2 Fig). The only time at which significant vaccine-derived reads were evident was in quadriceps muscles at 12 hours post vaccination (Fig 1A and S1A and S1B Table), suggesting that the vaccine had largely been cleared from the injection site by day 7. This is consistent with *in vitro* data showing that SCV induces cytopathic

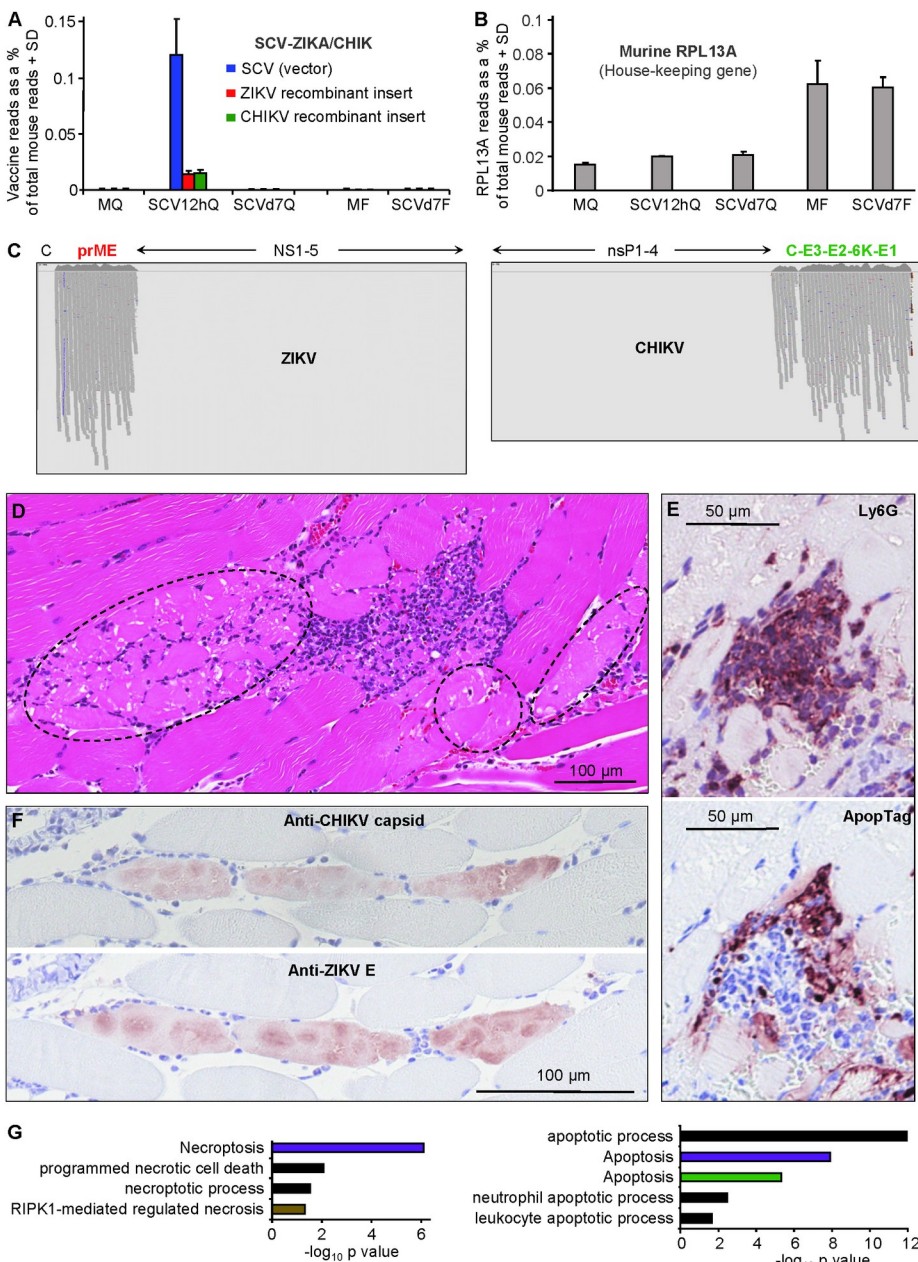

**Fig 1. Vaccine genome read alignments, histology and immunohistochemistry.** (A) RNA-Seq reads from each of the five groups aligned to the three viral genomes (the vector, SCV, and the two recombinant immunogen inserts from ZIKV and CHIKV); MQ—quadriceps muscles from mock vaccinated, SCV12hQ mice quadriceps muscles from SCV-ZIKA/CHIK vaccinated mice 12 hours post vaccination, SCVd7Q -quadriceps muscles from SCV-ZIKA/CHIK vaccinated mice taken 7 days post vaccination, MF–feet from mock vaccinated mice 7 days post vaccination, and SCVd7F - feet from SCV-ZIKA/CHIK vaccinated mice 7 days post vaccination. The number of viral reads is expressed as a percentage of the number of reads mapping to the mouse genome, with 3 biological replicates providing the SD (S1 Fig). The bars plotting to ≈0% had values ranging from 0 to $3.5\times10^{-5}$%. (B) RNA-Seq reads from each of the five groups aligned to the house-keeping gene, RPL13A, also expressed as a percentage of the number of reads mapping to the mouse genome. (C) IGV visualization of reads aligned to the recombinant structural polyprotein immunogens of ZIKV (prME) and CHIKV (C-E3-E2-6K-E1), which are encoded in the SCV-ZIKA/CHIK vaccine. All reads from all replicates are shown (for details see (S1A Table). As expected, no reads mapped to the non-structural genes of ZIKV or CHIKV (NS1-5 and nsP1-4, respectively), as these are not encoded in SCV-ZIKA/CHIK. (Vertical purple lines for ZIKV indicate base call errors after a string of Gs). (Reads mapping to the SCV genome are shown in S1B Table). (D) H&E staining of injection site 12 hours post vaccination. Dotted ovals indicate muscle cells in early stages of necrosis (pink staining). (E) Top; IHC with anti-Ly6G staining for neutrophils (parallel section to D focusing on area of

infiltrates). Bottom: Apoptag staining of the same area, illustrating apoptosis within areas of infiltrating cells. (F) Top: IHC for CHIK capsid protein. Bottom: parallel section showing IHC for ZIKA E protein. (G) Cell death annotation from Cytoscape analysis of up-regulated DEGs at 12 hours post vaccination (MQ vs SCV12hQ) (S2C Table) divided into non-apoptotic signatures (left) and apoptotic signatures (right). KEGG Pathways (purple), Go process (black), Reactome Pathways (brown), UniProt Keywords (green).

effects (CPE) in infected cells within a few days and that SCV is unable to produce viral progeny [1,18,19]. The data is also consistent with studies on MVA, where luciferase expression by MVA was lost 48 hours post inoculation [43]. The paucity of vaccine reads in the feet 7 days post vaccination (Fig 1A, SCVd7Q) also illustrates that the vaccine does not disseminate to and/or persist in joint tissues (a potential safety concern; see below). The percentage of reads mapping to a murine house-keeping gene, RPL13A [44], was similar for the 3 samples from quadriceps muscles, and for the two samples from feet (Fig 1B), illustrating that the low vaccine read counts for mock and day 7 samples (Fig 1A) were not due to low read counts for those samples.

A criticism of virally vectored vaccines has been that viral vector transcripts can be markedly more abundant than recombinant immunogen transcripts, resulting in immune responses excessively directed towards vector proteins rather than the recombinant immunogen(s) [45–47]. However, ≈20% of all the SCV-ZIKA/CHIK vaccine reads mapped to the two recombinant immunogen genes, even though the ZIKV and CHIKV sequences were relatively small (2067 bp and 3747 bp, respectively), when compared to the large SCV genome (≈190,000 bp). This perhaps attests to the strength of the poxvirus synthetic strong early late promoter [20] used for the CHIKV and ZIKV immunogens in the SCV-ZIKA/CHIK vaccine [18].

Expression of two immunogens in a single poxvirus vector construct carries the risk that one immunogen is expressed significantly better than the other, a problem encountered in a variety of settings [1]. A comparable number of reads mapped to the recombinant CHIK and ZIKA inserts (Fig 1A), with these two inserts distantly separate from each other in the SCV genome and driven from the same promoter [1]. This approach would seem largely to ensure (at least in SCV-ZIKA/CHIK) that comparable levels of mRNA are produced for each of the two immunogens.

Reads aligned to the CHIKV and ZIKV genomes were viewed using Integrative Genome Viewer (IGV) [48]. As expected, reads mapped to prME and C-E3-E2-6K-E1, which are encoded by SCV-ZIKA/CHIK; but not ZIKV capsid nor the non-structural proteins from both arboviruses (NS1-5 and nsP1-4), which are not encoded by SCV-ZIKA/CHIK (Fig 1C). Premature immunogen termination has been described previously for a VACV-based vaccine [49], with VACV transcription occurring in the cytoplasm [50]. No evidence for premature termination of SCV-ZIKA/CHIK immunogen transcription was apparent (Fig 1C).

Read alignments to genes encoded by SCV are described and annotated in detail in S1B Table. Immune and cell-death modulating proteins are highlighted, along with annotations regarding their activity in mice and their activity in the Copenhagen strain of VACV, from which SCV was derived. Many of these genes are referred to below.

## Injection site histology and immunohistochemistry at 12 hours post vaccination

H&E staining of the intramuscular injection sites showed that some skeletal muscle cells displayed fragmented pale cytoplasm with loss of striation and small condensed pyknotic nuclei indicative of necrosis (Fig 1D, dotted ovals); an enlarged image is shown in S3A Fig These

necrotic cells were partially surrounded by mixed inflammatory cells infiltrates (high densities of purple nuclei) and some cellular debris (Fig 1D). Immunohistochemistry (IHC) with a neutrophil-specific marker, anti-Ly6G [51,52], illustrated that the infiltrates contained abundant neutrophils (Fig 1E, top panel, Ly6G). Interestingly, neutrophils have been shown to contribute to adjuvant activity [53]. The infiltrates also contained areas staining with ApopTag indicating apoptosis (Fig 1E, parallel section, bottom panel, ApopTag).

IHC with monoclonal antibodies recognizing CHIKV capsid (5.5G9 [54] and ZIKV envelope (4G2 [55], clearly illustrated expression of vaccine antigens in skeletal muscle cells 12 hours post infection (SCV12hQ) (Fig 1F). Using parallel sections, IHC with 4G2 (S3B Fig) and a control antibody is shown in S3C Fig, with no significant staining observed in the latter. The spherical/oval cytoplasmic staining patterns (Fig 1F) likely reflect the well described cytoplasmic factories wherein the poxvirus coordinates protein expression and subjugates host functions [56–58]. No significant staining was observed in MQ. These results illustrated that the immunogen mRNA expression seen in Fig 1A and 1C translates into protein expression in muscle cells *in vivo*.

## Injection site host cell death signatures at 12 hours post vaccination

The mode of cell death for a host cell expressing vaccine immunogens can have important implications for immunogenicity, with necrosis often favored over apoptosis [14,59]. RNA-Seq analysis of the mouse i.m. injection sites 12 hours post vaccination (MQ vs SCV12hQ; full gene list in S2A Table) provided a set of differentially expressed genes (DEGs) (S2B Table; FDR or q <0.01, fold change >2 and sum of all counts across the six samples >6). The up-regulated DEGs (n = 1390; S2B Table) were analyzed by Cytoscape (S2C Table), with cell death terms suggesting the presence of apoptosis, necroptosis and necrosis (Fig 1G). Skeletal muscle cells are generally resistant to apoptosis [60,61] and VACV's apoptosis inhibitor, B13R [14], was also expressed at the injection site (S1B Table). Skeletal muscle cells have recently been shown to be able to undergo necroptosis [62]. Skeletal muscle necrosis is well described [63,64] and H&E staining was consistent with muscle cell necrosis (Figs 1D and S3A). As ApopTag staining was clearly present in the aforementioned infiltrates (Fig 1E, ApopTag), the apoptosis signatures (Fig 1G) may largely be associated with infiltrating leukocytes such as neutrophils, which are highly prone to apoptosis [65]. MVA can induce apoptosis *in vitro* and in certain settings *in vivo* [14,66] and SCV can induce apoptosis (at least *in vitro*, S4 Fig); however, the mode of cell death elucidated *in vitro* may not be recapitulated in primary skeletal muscle cells *in vivo*.

## Large Toll-like receptor signatures at 12 hours post vaccination

The up-regulated DEGs (for MQ vs SCV12hQ; S2B Table) analyzed as above by Cytoscape returned multiple terms associated with innate immune responses (S2C Table). To provide insights into the early innate host immune responses and potential adjuvant signatures induced by SCV-ZIKA/CHIK vaccination, the full DEG list (1608 genes) for MQ vs SCV12hQ (S2B Table) was analyzed by Ingenuity Pathway Analysis (IPA) using the Up-Stream Regulator (USR) function and the direct and indirect interaction option. The list of USRs (S2D Table) illustrated a highly significant Toll-like receptor (TLR) signature, dominated by TLR3 and 4, followed by TLR9, 7 and 2 (Fig 2A). Although other TLRs (TLR1, 5, 6, 7/8, 8) were also identified, the number of unique DEGs responsible for these annotations was low (Fig 2A, numbers in brackets), arguing that these were less reliable USRs as they arose from subsets of DEGs already used in the annotations for TLR3, 4, 9, 7 and/or 2 (S2D Table, Target molecules in dataset). Given the common signaling pathways used by all TLRs, primarily involving MyD88 and/or TICAM1/TRIF, overlap in genes induced via the different TLRs would be expected.

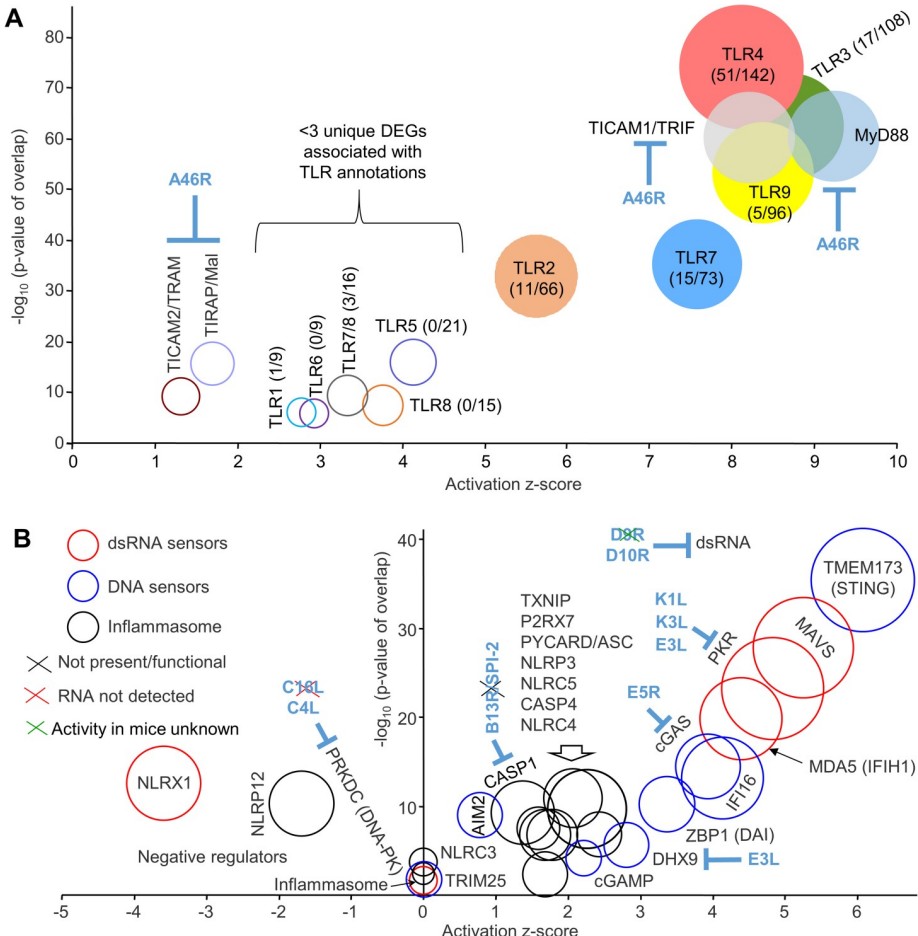

**Fig 2. TLR and cytosolic sensor signatures at 12 hours post vaccination.** (A) TLR signatures identified by IPA USR analysis (S2D Table) of 1608 DEGs identified in quadriceps muscles 12 hours post SCV-ZIKA/CHIK vaccination (MQ vs SCV12hQ; S2B Table). Circle diameters reflect the number of DEGs associated with each IPA USR annotation. Numbers in brackets indicate the number of unique DEGs associated with each annotation over the total number of DEGs associated with the TLR annotation; circles with colored fills contain >3 DEGs uniquely associated with the indicated TLR annotation. A46R is expressed in the cytoplasm of infected cells. (B) Cytosolic sensor signatures identified by IPA USR analysis (S2D Table). Sensors divided into 3 categories associated with dsRNA (red circles), DNA (blue circles) and inflammasome activation (black circles). Circle diameters reflect the number of DEGs associated with each annotation. VACV genes encoding cytoplasmic inhibitors are shown in blue, with the black cross indicating that the gene/protein is not present or not functional in SCV (or in the Copenhagen strain of VACV), the red cross indicating that the gene was not detected by RNA-Seq of MQ vs SCV12hQ, the green cross indicate that the activity in mice is unknown (see S1B Table).

The z-scores and p values for TLR signatures determined herein (Fig 2A) showed a remarkable concordance with previously published TLR-knockout mouse studies (summarized in S2E Table). The higher the z-score/p value, the more important the TLR was for infection and pathology in TLR-knockout mice infected with replication competent poxviruses. Specifically, the top TLR, TLR4, is stimulated by an unknown ligand present in/on VACV particles, with TLR4 required for effective antiviral activity and protection against mortality in mice after VACV infection [67]. TLR3 stimulation is likely mediated by dsRNA derived from the abundant complementary RNA transcripts produced late in the VACV infection cycle [68]. TLR3 stimulation in VACV-infected mice promotes inflammatory cytokine production, immunopathology, and affects mortality [69]. TLR7 (which detects ssRNA) is expressed on plasmacytoid

dendritic cells and B cells, with TLR7 and TLR9 important for type I interferon secretion by dendritic cells following fowlpox infection [70]. TLR9 is required for survival of mice following ectromelia virus infections [71,72] and is likely stimulated by viral unmethylated ssDNA containing CpG motifs [73] and/or mitochondrial DNA [74] released by viral CPE. TLR2 stimulation during VACV infection in mice has minimal impact on viral replication [75], but does promote NK activation and CD8 T cell expansion and memory [76,77]. Thus both SCV and VACV would appear to stimulate TLR2, whereas MVA is reported not to do so [78]. To the best of our knowledge, there is no literature suggesting a role for TLR1, 5 or 6 in poxvirus infections, consistent with the low number of unique DEGs for these annotations (Fig 2A). The role of TLR8 in VACV infections remains controversial [79], with TLR8 non-functional in mice [80].

VACV produces an inhibitor of TRIF, MYD88, TRAM and MAL, called A46 or VIPER (encoded by A46R), a protein reported to be active in murine systems [81]. However, A46 is expressed in the cytoplasm of SCV-infected cells and not in neighboring uninfected cells that may also express TLRs. Such cells might sense TLR agonists comprising viral pathogen-associated molecular patterns (PAMPs) and/or damage-associated molecular patterns (DAMPs) released by SCV infection-induced CPE [19,82,83].

## Multiple cytoplasmic sensor signatures at 12 hours post vaccination

The IPA analysis of DEGs for MQ vs SCV12hQ (S2D Table, direct and indirect) produced a series of USRs associated with (i) detection of cytoplasmic dsRNA (Fig 2B, red circles) via MAVS, MDA5 and PKR, (ii) detection of cytoplasmic DNA (Fig 2B, blue circles) dominated by STING/IFI16/cGAS, and (iii) activation of the inflammasome (Fig 2B, black circles). These results (as for TLRs) again recapitulated the relative importance of these cytoplasmic sensors observed during the full course of infection of knockout mice with replication competent poxviruses (summarized in S2E Table; see Discussion). dsRNA from complementary VACV RNA transcripts stimulates MAVS signaling [84], likely via MDA5 [85]. Stimulation of MDA5 or RIG-I and PKR by VACV *in vitro* has been reported previously [84,86], with both MAVS and MDA5 reported to contribute to host defense against VACV infection [85]. PKR activation is also enhanced by MDA5 [87]. Like SCV, the canarypox virus vector, ALVAC, also stimulates the cGAS/IFI16/STING pathway [88]. Activation of the proteases Caspase 1 (gene CASP1) (canonical) and Caspase 11 (gene CASP4) (non-canonical) represent the central outcomes of inflammasome activation, with VACV stimulation of the inflammasome well described [89]. ALVAC is also reported to stimulate the inflammasome via AIM2 in both human and mouse cells [88], with a minor AIM2 signature also seen after SCV-ZIKA/CHIK (Fig 2B). Viron assembly is arrested at the viroplasma stage in SCV-infected host cells due to the deletion of D13L [19], which may limit inflammasome activation.

Poxviruses encode a number of proteins that seek to limit the activity of host immune responses (S1B Table, yellow highlighting), with some of these inhibiting the activities of cytoplasmic sensors (Fig 2B, blue text). VACV's decapping enzymes (D9 and D10, encoded by D9R and D10R) are expressed at the vaccination site (S1B Table). Both proteins inhibit dsRNA accumulation, with D10 functional in mice [90]; whether D9 is functional in mice is unknown (Fig 2B, green cross). DHX9 is involved in both DNA and RNA sensing and is targeted by VACV's E3 protein (encoded by E3L), with PKR inhibition the best defined activity of E3 [91–93]. VACV DNA is usually shielded from cytoplasmic sensors during replication in viral factories via wrapping in ER membranes; however, this wrapping is lost during virion assembly [94]. The DNA sensor PRKDC/DNA-PK (a DNA-dependent protein kinase) showed a z-score of zero (Fig 2B), perhaps due to the inhibitory activity of C4 (a protein encoded by C4L) [95].

The expression of PRKDC/DNA-PK mRNA was not significantly altered (S2D Table), consistent with C4 protein-protein interactions [95] inhibiting the transcriptional modulation mediated by this upstream regulator. Transcripts for another PRKDC/DNA-PK inhibitor, C16L, were not detected by RNA-Seq in SCV12hQ (Fig 2B, red cross and S1B Table). CrmA (from cowpox) and VACV's homologue, B13R/SPI-2, inhibit caspase 1 (and other caspases), but are not functional in the Copenhagen strain of VACV [96] (Fig 2B, black cross). NLRP1 was not identified by the IPA USR analysis, potentially due to the expression of F1L (S1B Table) [97].

## Dominant TLR-signaling associated signatures at 12 hours post vaccination

Following stimulation of TLR4, 9, 7 and 2 (but not TLR3) (Fig 2A), a series of signaling events are initiated via the Myddosome, which contains MyD88/IRAK2/IRAK4 and signals to IRAK1 and TRAF6, with TRAF3 acting as a negative regulator [98,99]. TRAF5 [100], TRAFD1/ FLN29 [101] and IRAK3 (aka IRAK-M) [102] are also negative regulators of TLR signaling. TLR3 signaling also involves TRAF6 and TRAF3. All the aforementioned signaling molecules were identified by the IPA USR analysis, with negative regulators having negative z-scores and the rest positive z-scores (Fig 3A and S2D Table, direct and indirect). These results are entirely consistent with the dominant TLR signatures illustrated in Fig 2A. IL-1 receptor signaling also involves many of the same signaling molecules as TLR signaling [103], with IL-1β a major cytokine USR (see below). The dominance of TBK1 may reflect its involvement in a series of signaling pathways; specifically, TLRs (including TLR3), STING and MDA5/MAVS [104] that were illustrated in Fig 2. C6 (encoded by C6L) inhibits TBK1 via binding to TBK1 adaptors (such as TANK) (Fig 3A) thereby inhibiting activation of IRF3 and IRF7 [105,106]. K7 (encoded by K7R) binds DDX3 [107], an adaptor protein for the TBK1/IKKε complex that promotes IRF3 phosphorylation [105,108]. TLR signaling is also inhibited by K7 and A52 (encoded by A52R), which bind to IRAK2 and TRAF6 [96,109,110]. N1 (encoded by N1L) binds TBK1 and inhibits NF-κB and IRF3 signaling pathways [111–113] (Fig 3A).

## Interferon response factor signatures at 12 hours post vaccination

Interferon response factors (IRFs) are key transcription factors triggered by PAMPs, with IRF3, IRF7 and IRF1 dominating (Fig 3B) in the IPA USR analysis (S2D Table, direct and indirect) of the DEGs from MQ vs SCV12hQ (S2B Table). These 3 IRFs are also in the top 5 USRs (sorted by p value) when the "direct" only option was used for the IPA USR analysis (S2D Table, direct only). IRF3 and IRF7 are activated via multiple PAMP sensors described in Fig 2 [87,104,114], are intimately involved in driving antiviral responses [115–117], are also activated by MVA [111] and ALVAC [118], and have been shown to promote adaptive immune responses in a number of settings [119–121]. SCV also encodes N2L, with N2 inhibiting IRF3 activation in poxvirus infected cells [96]. IRF1 has a role in positive feedback maintenance of ISG expression [116,122] and has been shown to promote adaptive immunity in certain settings [123,124].

Other IRF USRs included IRF2, 4, 5, 6, 8 and 9 (Fig 3B). IRF8 has a critical role in development and maturation of myeloid cells such as dendritic cells [125] and IRF5 is predominantly expressed by myeloid cells and regulates inflammatory responses, generally downstream of TLR-MyD88 pathways [126]. IRF4 has a negative z-score (Fig 3B) and is a negative regulator of TLR signaling [127].

## Canonical NF-κB family signatures at 12 hours post vaccination

The NF-κB family of transcription factors play key roles in immunity, with the IPA USR analysis (S2D Table, direct and indirect) illustrating a dominant canonical NF-κB signature at the

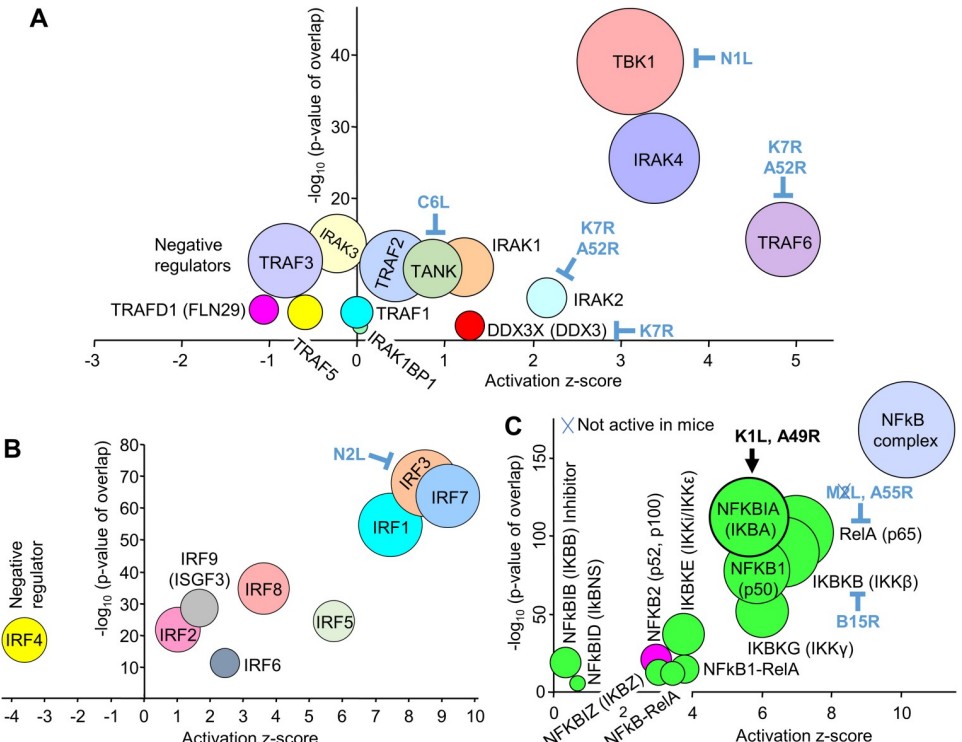

**Fig 3. Secondary messenger signatures at 12 hours post vaccination.** (A) Secondary messenger signatures. VACV encodes a series of cytoplasmic inhibitors, which are indicated in blue text. (B) Interferon response factors (IRFs). (C) NF-κB signatures. Green fill indicates canonical pathways, magenta fill non-canonical pathway, blue fill—not assigned to canonical or non-canonical. VACV-encoded cytoplasmic inhibitors are shown in blue text. K1L and A49R (black text) enhance the activity of NFKBIA, an inhibitor of the canonical pathway. Blue cross means the inhibitor is not active in mice.

injection site (Fig 3C, green circles), consistent with the TLR signaling USRs described above. The dominance of the NFKBIA, but not another NF-κB inhibitor NFkBIB, may reflect the activities of the K1L and A49R genes in vaccine-infected cells. Both K1 and A49 proteins prevent degradation of NFKBIA [96], with A49 binding the ubiquitin ligase B-TrCP [128]. B14 (encoded by B15R in the Copenhagen strain) binds and inhibits IKBKB [129], and intracellular M2 (encoded by M2L) inhibits RelA (p65) nuclear translocation, but is not active in mice [96] (Fig 3C, blue cross). A55 (encoded by A55R) dysregulates NF-κB signaling by disrupting p65-importin interaction, is active in mice [130] and is expressed by SCV at the injection site (S1B Table). As the SCV-encoded inhibitors of NF-κB signaling are expressed only in the SCV-infected cells, the dominant NF-κB signatures are likely largely associated with uninfected cells stimulated and/or recruited by the pro-inflammatory environment [131].

## Th1 cytokine signatures at 12 hours post vaccination

The cytokine USR profile at 12 hours post vaccination is dominated by cytokine signatures generally associated with Th1 responses (Fig 4A, red circles), in particular TNF, IL-1β and IFNγ, with *in vivo* induction of these cytokines by VACV suggested by previous studies [36,132,133]. This Th1 dominance is consistent with studies on recombinant MVA vaccines [134,135]. TNF is required for optimal adaptive immune responses to VACV and other immunogens [36,136]. IL-1 is important for host immune responses to VACV [132], with many vaccine adjuvants also inducing the release of IL-1 [137]. Finally, IFNγ has anti-VACV activity

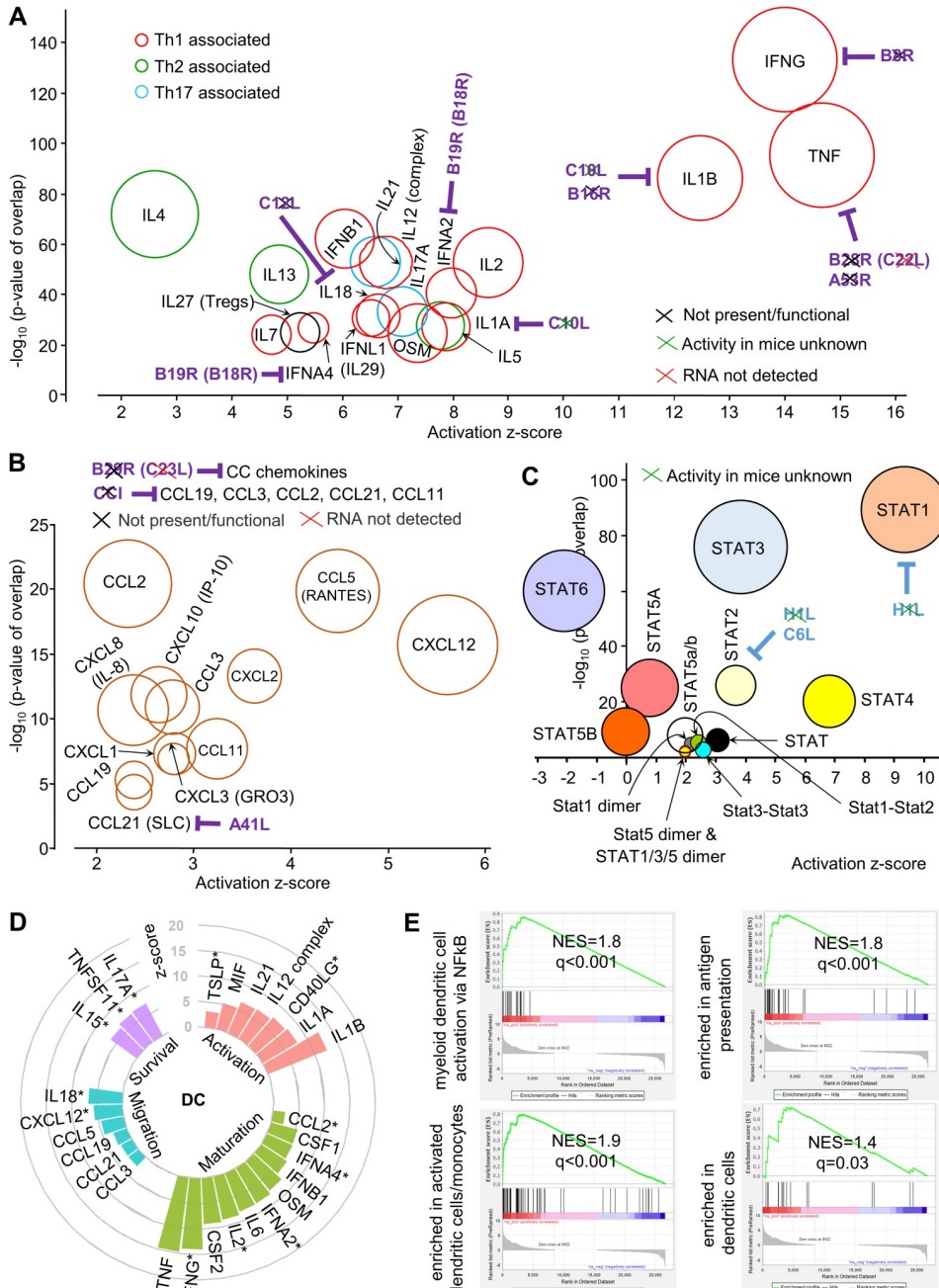

**Fig 4. Cytokine, chemokine, dendritic cell and STAT signatures at 12 hours post vaccination.** (A) Cytokine signatures. VACV genes encoding secreted inhibitors are shown in purple text. Black crosses indicate the inhibitors are not present or functional in SCV. Green crosses indicate that the activity in mice is unknown. Red crosses indicate the RNA was not detected in our RNA-Seq analysis. (B) Chemokine signatures. Purple text and crosses as in A. (C) STAT signatures. VACV encoded cytoplasmic inhibitors are shown in blue text. Crosses as in A. (D) IPA USRs associated with stimulation of dendritic cells. * indicates that the mediators have more than one of the four dendritic activities indicated. (The figure includes some USRs present in previous bubble graphs). For references see S2F Table. (E) GSEAs for the Blood Transcript Modules (right to left) M43.0 and M43.1 (gene sets combined, n = 21); M95.0, M95.1, M71 and M200 (gene sets combined, n = 49); M64, M67, M119 and M165) (gene sets combined, n = 71); M168 (n = 19). For gene set details see S2F Table.

[138] and has adjuvant properties in a range of settings [139–141]. Although IL-27 was initially associated with Th1 responses, it is now recognized as a promoter of T regulatory cells [142] (Fig 4A, IL27).

VACV encodes a number of soluble inhibitors of several cytokines (Fig 4A, purple text). C10L encodes C10, which blocks interaction of IL-1 with its receptor [143], but its activity in mice is unknown (Fig 4A, green cross). B16 (encoded by B16R) (B15R in Western Reserve) is a secreted IL-1β decoy receptor [144,145], but appears to be truncated in the Copenhagen strain of VACV (Uniprot; GCA 006458465.1). B8 (encoded by B8R) is a secreted IFNγ receptor homologue, and ZIKA prME was inserted into the B7R-B8R locus in SCV-ZIKA/CHIK thereby inactivating these genes [18]. B28R (C22L) and A53R encode TNF receptor homologues that are not active in the Copenhagen strain of VACV [146]. C12 (encoded by C12L) inhibits IL-18 [147]. B19 (encoded by B19R) (also known as B18R in other VACV strains) is a decoy receptor for soluble IFNαs [148,149]. B19 mRNA is well expressed at the injection site (S1B Table) and is potentially responsible for the relatively low z-scores of the IFNα USRs (Fig 4A). A35 (encoded by A35R) (not shown in the figures) is an intracellular VACV protein that also inhibits the synthesis of a number of chemokines and cytokines (including IFNα, MIP1α, IL-1β, IL-1α, GM-CSF, IL-2, IL-17, GRO1/KC/CXCL1, RANTES, TNFα) by VACV-infected cells [150].

## Chemokine signatures at 12 hours post vaccination

The chemokine signatures (Fig 4B) are dominated by (i) CXCL12, which is made by many cell types and is strongly chemotactic for lymphocytes, (ii) CCL5 (RANTES), which is *inter alia* chemotactic for T cells and (iii) CXCL2, a neutrophil chemoattractant (consistent with Fig 1E). CXCL12 and CCL5 are also involved in dendritic cell (DC) recruitment [151,152]. CCL2 (Fig 4B) is also induced by MVA [153,154] and is involved in DC maturation and induction of T cell immunity [155].

VACV infected cells secrete a number of proteins that bind and inhibit certain chemokines, although only A41 (encoded A41L) is active in the Copenhagen strain of vaccinia (Fig 4B, purple text). A41 inhibits CCL21 [96], perhaps consistent with its low z-score (Fig 4B). B29, encoded by B29R (C23L) inhibits multiple CC chemokines, but is inactive in the Copenhagen strain [156]. CCI inhibits a series of chemokines (Fig 4B), but is not expressed on the Copenhagen strain of VACV [157,158].

## STAT signatures at 12 hours post vaccination

Cytokines and chemokines bind to their receptors and activate transcription via STATs. The dominant STAT signatures (from S2D Table, IPA direct and indirect) were STAT1, STAT4 and STAT3 (Fig 4C), with STAT1 and STAT3 representing the top USRs by p value and STAT1 also the top USR by z-score when analyzed by IPA using direct only interaction (S2D Table, direct only). STAT1 forms complexes primarily STAT1-STAT1 homodimers (stimulated by IFNγ signaling) and STAT1-STAT2-IRF9 (ISGF3) (stimulated by type I IFN signaling). Using Interferome to interrogate the "Target molecules in dataset" listed for the STAT1 signature (S2D Table, direct and indirect), nearly all the target molecules were deemed IFNγ inducible (with most also inducible by type I IFNs). The relatively low p values and z-scores for STAT1 dimers would thus appear to be an under-annotation within IPA. The dominant STAT1 signature (Fig 4C) is consistent with the dominant IFNγ signature in Fig 4A. The cytoplasmic VACV-expressed H1 (encoded by H1L) inhibits STAT1 and STAT2 [159], but again only in cells infected with VACV. C6 (encoded by C6L), as well as the aforementioned binding of TBK1 adaptors, also binds the TAD domain of STAT2 [160].

STAT4 signaling is induced by a number of cytokines including IL-12 and IL-2 [161] and is critical for IFNγ production during generation of Th1 responses [162]. STAT3 signaling is induced by a number of cytokines including IL-6 and OSM (and growth factors such as GM-CSF), with BCG vaccination recently shown to cause STAT3 phosphorylation in antigen presenting cells [163]. STAT6 is involved in driving Th2 responses [164] and has a negative z-score (Fig 4C), consistent with the Th1 dominance illustrated in Fig 4A.

### Dendritic cell associated signatures

A range of mediators affect dendritic cell activities, with many of these identified as USRs by IPA analysis of DEGs for MQ vs SCV12hQ (Fig 4D andS2D Table; for references see S2F Table). The VACV protein A35 inhibits a number of these mediators (see above), as well as inhibiting class II antigen presentation [150]. Multiple key mediators needed for induction of adaptive immune responses by dendritic cells would thus appear to be active at the injection site 12 hours post vaccination.

Extensive bioinformatics treatments of >30,000 peripheral blood transcriptomes from >500 human studies of 5 vaccines provided 334 publically available gene sets in the form of Blood Transcription Modules (BTMs). BTM gene sets are associated with specific subsets of cells and/or their activities [165]. BTM gene sets associated with dendritic cells and dendritic cell activities (S2F Table) and Gene Set Enrichment Analyses (GSEAs) were used to determine whether genes from dendritic cell BTMs were significantly represented in the MQ vs SCV12hQ gene list (S2A Table). The GSEAs provided highly significant results (Fig 4E), illustrating that signatures associated with dendritic cells and their activities can be readily identified at the injection site 12 hours post vaccination. Such signatures likely underpin the immunogenicity of the poxvector system.

### Injection site signatures at day 7 post vaccination

The most common side effects reported for MVA (licensed as a small pox vaccine in Europe as IMVANEX) were at the site of subcutaneous injection; most of them were mild to moderate in nature and resolved without any treatment within seven days [166]. To gain insights into the injection site responses after SCV vaccination, RNA-Seq of muscles on day 7 post-vaccination was undertaken to provide a gene list (MQ vs SCVd7Q, S2G Table), from which a DEG list (n = 1413 genes) was generated (S2H Table) by applying the same filters as above (q <0.01, FC >2 and sum of all counts across the six samples >6). Of the 1413 DEGs, 1337 were up-regulated, with 633 (47%) of these also up-regulated DEGs for MQ vs SCV12hQ.

Cytoscape analyses of up-regulated DEGs from MQ vs SCV12hQ (S2C Table) were compared with MQ vs SCVd7Q (S2I Table). Multiple top signatures (by FDR) associated with T cells and B cells were substantially more significant on day 7 than at 12 hours (Fig 5A and S2J Table). For instance, FDR values associated with the GO Process terms "positive regulation of T cell activation" and "T cell differentiation" were ≈9 logs more significant by day 7, when compared with 12 hours post vaccination (Fig 5A and S2J Table). T cell receptor associated KEGG Pathways and GO Component terms were also more significant on day 7 (Fig 5A and S2J Table). "T cell receptor complex" was also the top "GO Cellular Component" term by p value for day 7 up-regulated DEGs (S2J Table, Enrichr). A similar pattern emerged for B cell terms (Fig 5A and S2J Table).

GSEAs (as in Fig 4E) using gene lists from BTMs [165] associated with T cell differentiation and division, and B cell differentiation into plasma cells, showed significance for SCVd7Q, but not SCV12hQ (Fig 5B). Thus remarkably, these BTMs were able to identify signatures at the

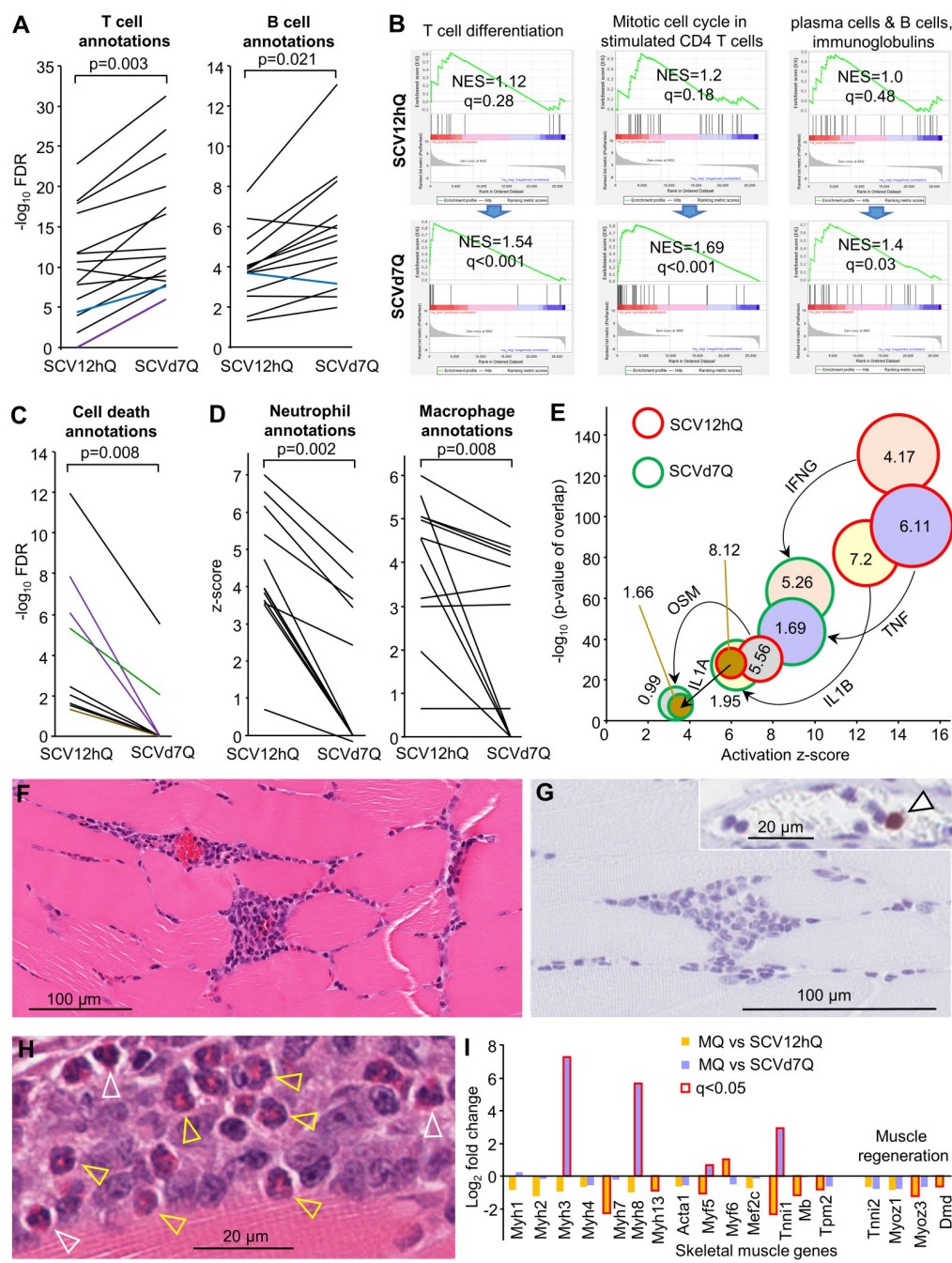

**Fig 5. The injection site day 7 post vaccination.** (A) Cytoscape analysis of up-regulated DEGs from MQ vs SCV12hQ and MQ vs SCVd7Q, illustrating the upward trend in significance of top B and T cell associated annotations. Black line–GO Process, Blue line GO—Component, Purple line—KEGG Pathways. For full lists and descriptions of annotations see S2J Table; statistics by paired t test for the full lists (parametric data distribution). (B) GSEAs were used to interrogate MQ vs SCV12hQ and MQ vs SCVd7Q gene lists using T cell and B cell BTMs (left to right) M14 (n = 12); M4.5 (n = 35); M156.0 and M156.1 (gene sets combined, n = 56) (for details of BTM gene see S2J Table). (C) As for A illustrating the downward trend of cell death annotations. Anotations not identified by the IPA analysis were nominally given a -log₁₀ FDR value of zero (y axis). Color coding as for A, but also Green line–UniProt Keywords, Brown line–Reactome Pathways. For descriptions of annotations see S2K Table. (Statistics by Wilcoxon Signed Rank tests; non-parametric data distribution). (D) IPA *Diseases and Functions* analysis of DEGs (up and down-regulated) from MQ vs SCV12hQ and MQ vs SCVd7Q, illustrating the downward trend in z-scores for macrophage and neutrophil annotations (for description of annotations see S2L Table and S2M Table). (Statistics by Wilcoxon Signed Rank tests). (E) Major IPA USR pro-inflammatory cytokine annotations identified at 12 hours (Fig 4A and S2D

Table) had much lower z-scores and p values on day 7 post vaccination (S2N Table). Numbers in the circles represent the $\log_2$ fold change for that cytokine relative to MQ. (F) H&E staining of intramuscular injection site lesions on day 7 post infection. (G) Neutrophil Ly6G staining of lesions from day 7 post infection. Arrow in insert shows positive staining of a neutrophil in a blood vessel capillary. (H) Eosinophils in the intramuscular injection site lesions on day 7 post vaccination. White arrow heads—mature segmented eosinophils. Yellow arrowheads–immature band eosinophils. (I) Expression of skeletal muscle genes from MQ vs SCV12hQ (S2A Table) and MQ vs SCVd7Q (S2G Table); bars with red outline indicate significant fold change (q<0.05).

injection site on day 7 that were associated with the development of adaptive immune responses. (IgG responses are known to be induced after SCV-ZIKA/CHIK vaccination [18]).

Vaccination site lesions are well described for VACV vaccination [31], with skin lesions reported days 6–11 after vaccination with the Lister strain [32] and days 3–19 after Dryvax vaccination [33]. Such lesions are associated with cell death [34], tissue damage [35] and recruitment of neutrophils, with neutrophil recruitment also a feature of *eczema vaccinatum*, a complication of smallpox vaccination [31,36,37]. Cytoscape analyses (S2C Table and S2I Table) illustrated that the cell death pathway annotations identified at 12 hours (Fig 1G) were considerably less significant or absent for day 7 (Fig 5C and S2K Table). Analysis of the 1413 DEGs from MQ vs SCVd7Q (S2H Table) with IPA *Diseases and Functions* feature (S2L Table), showed a significant reduction in the z-scores of neutrophil-associated annotations on day 7 when compared to 12 hours (Fig 5D and S2M Table). (The Cytoscape analysis also showed a highly significant reduction in FDR values for neutrophil terms, S2I Table, graph on right). These analyses indicate that progression of cell death and neutrophil infiltration is not a feature of SCV vaccination, likely consistent with the inability of SCV to produce viral progeny [19]. SCV does not cause a spreading infection, with vaccine-derived mRNA lost by day 7 (Fig 1A). A similar IPA *Diseases and Functions* analysis of macrophage-associated annotations also illustrated a significant reduction by z-scores (Fig 5D and S2M Table), further indicating that injection site inflammatory responses were abating by day 7 [65].

The dominant pro-inflammatory cytokine USRs identified at 12 hours post vaccination (Fig 4A) were substantially lower by day 7 post vaccination with respect to both–$\log_{10}$ p values and z-scores (Fig 5E and S2D Table vs S2N Table). Fold changes in cytokine mRNA expression levels relative to MQ were also substantially lower on day 7 (Fig 5E), with the exception of IFNγ, which had a fold change relative to MQ of 4.17 at 12 h and a fold change of 5.26 relative to MQ on day 7, perhaps due to the emerging Th1 T cell responses (see above). IPA *Diseases and Functions* also showed reduced significance and z scores on day 7 for *Inflammatory response* (-$\log_{10}$ p value 110.7 to 59.8, z-score 7.7 to 5.5) and *Chronic inflammatory disorder* (-$\log_{10}$ p value 70.1 to 38.1, z-score -0.23 to -2.2) (S2L Table). These analyses again argue that inflammation at the injection site is abating on day 7, with persistent inflammation at the injection site generally deemed undesirable in most vaccination settings [167–169].

## Loss of neutrophils and presence of eosinophils on day 7 post vaccination

H&E staining of the injection sites day 7 post vaccination supports the bioinformatics results described in the previous section. When compared with 12 hours (Fig 1D), necrotic muscle lesions were largely absent, with the cellular infiltrates less disseminated and more focal (Fig 5F). In addition, in contrast to 12 hours (Fig 1E), neutrophils (stained with anti-Ly6G) were not observed in the day 7 cellular infiltrates (Fig 5G), although the occasional neutrophil could be seen in blood vessels, illustrating that the staining had worked (Fig 5G, insert, arrowhead). In contrast to Fig 1E, ApopTag staining was also largely negative on day 7 (not shown). Loss of neutrophils is consistent with inflammation resolution [65].

Another feature of the resolving infiltrates on day 7 post vaccination (clearly evident from H&E staining) was the presence of eosinophils (Fig 5H), despite the retention of a dominant Th1 signature (Fig 5E). Many of these cells showed the morphological features of immature band eosinophils, as distinct from segmented mature eosinophils (Fig 5H).

At 12 hours post vaccination, genes specific to skeletal muscle were generally slightly down-regulated (Fig 5I and S2A Table), consistent with the SCV infection-associated necrosis or pyroptosis (Fig 1D). On day 7 post vaccination, Myh3 and Myh8 were significantly up-regulated (Fig 5I and S2G Table), consistent with these genes being transiently up-regulated after muscle injury [170]. Tnni1, a skeletal myogenesis marker [171], was also up-regulated (Fig 5I). However, stable expression of Tnni2, Myoz1, Myoz3 and Dmd by day 7 (Fig 5I), argues that muscle regeneration had been largely completed at this time [170]; consistent with the H&E staining (Fig 5F).

## Concordance with vaccine and virus infection gene sets

The Molecular Signatures Database (MSigDB) provides a collection of >31,000 gene sets for use in GSEAs. All these gene sets were used to interrogate the full pre-ranked gene lists for MQ vs SCV12hQ (S2A Table) and MQ vs SCVd7Q (S2G Table). This analysis thus uses all the genes, rather than just the DEGs. Several signatures (with q<0.05) associated with influenza and yellow fever vaccines were identified (S5 Fig), illustrating that there are significant similarities in the gene expression profiles for SCV-ZIKA/CHIK and two licensed virus vaccines. Influenza and yellow fever vaccines are the only vaccines for which signatures are present in MSigDB. Multiple signatures (with q<0.05) were also identified that were associated with a range of virus infections and generalized anti-viral defense (S5 Fig), arguing that SCV-ZIKA/CHIK, although unable to generate viral progeny, nevertheless stimulates common shared anti-viral responses that are induced by multiple viruses.

## No detection of adventitious agents or microbial infections

The SCV-ZIKA/CHIK vaccine was tested for sterility and mycoplasma before inoculation. However, insights into unforeseen microbial contamination (adventitious agents) present in the vaccine preparation [172] or microbial infection(s) that might have been introduced during vaccination [173], can also be gleaned from injection-site vaccinology. The reads that were not mapped or assigned to the mouse (S1C Fig) or the vaccine genomes (Fig 1A) by the STAR aligner were thus analyzed by Kraken, a metagenomic sequence classification tool [174,175]. Contamination of RNA-Seq samples with RNA from various sources from the laboratory and the environment is a well-recognized phenomenon [176–178], so samples that did not receive the SCV vaccine (MQ) provided a baseline for such background contamination. Unmapped and unassigned reads from MQ primarily identified *Homo sapiens*, *Pasteurella* and murine retroviruses (S6 Fig). Almost identical metagenomic patterns emerged for unmapped/unassigned reads from SCV12hQ and SCVd7Q (S6 Fig), arguing against the presence of adventitious agents in the vaccine, or injection-site infections in SCV-vaccinated animals being responsible for the adjuvant signatures described above.

## No compelling arthritic signatures in feet on day 7 post vaccination

Arthralgia is a common (>1/100 to <1/10) acute transient adverse event in humans after vaccination with IMVANEX [38] or recombinant MVA vaccines [39,40]. To determine whether SCV-ZIKA/CHIK vaccination is associated with the induction of an arthritic signature on day 7 post vaccination, feet were collected and were analyzed by RNA-Seq (MF vs SCVd7F) (S2P Table). A DEG list was generated after application of two filters q<0.05 and the sum of counts

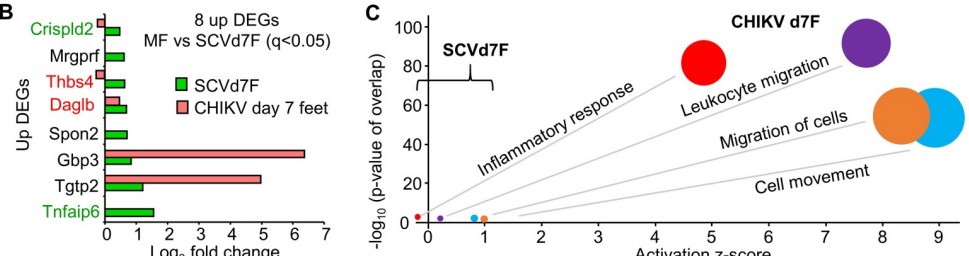

**A**

| Name of MSigDB gene set | SIZE | NES | NOM p-val | FDR q-val |
| --- | --- | --- | --- | --- |
| GSE22140_GERMFREE_VS_SPF_ARTHRITIC_MOUSE_CD4_TCELL_UP | 175 | 1.14 | 0.17 | 1.00 |
| HP_OSTEOARTHRITIS | 51 | 1.13 | 0.23 | 1.00 |
| GSE22140_HEALTHY_VS_ARTHRITIC_MOUSE_CD4_TCELL_UP | 172 | 1.10 | 0.23 | 1.00 |
| GSE22140_HEALTHY_VS_ARTHRITIC_GERMFREE_MOUSE_CD4_TCELL_DN | 183 | 1.10 | 0.21 | 1.00 |
| HP_ARTHRITIS | 164 | 0.88 | 0.79 | 1.00 |
| HP_HIP_OSTEOARTHRITIS | 9 | 0.83 | 0.71 | 1.00 |
| GSE10500_ARTHRITIC_SYNOVIAL_FLUID_VS_HEALTHY_MACROPHAGE_UP | 135 | 0.80 | 0.93 | 1.00 |
| GSE22140_HEALTHY_VS_ARTHRITIC_GERMFREE_MOUSE_CD4_TCELL_UP | 169 | 0.76 | 0.98 | 1.00 |
| HP_ARTHRALGIA | 135 | 0.67 | 1.00 | 1.00 |
| GSE22140_GERMFREE_VS_SPF_ARTHRITIC_MOUSE_CD4_TCELL_DN | 165 | 0.56 | 1.00 | 1.00 |
| GSE22140_HEALTHY_VS_ARTHRITIC_MOUSE_CD4_TCELL_DN | 182 | 0.56 | 1.00 | 0.99 |
| HP_RHEUMATOID_ARTHRITIS | 18 | -0.98 | 0.50 | 1.00 |
| GSE10500_ARTHRITIC_SYNOVIAL_FLUID_VS_HEALTHY_MACROPHAGE_DN | 134 | -0.94 | 0.60 | 1.00 |
| HP_POLYARTICULAR_ARTHROPATHY | 12 | -0.78 | 0.76 | 1.00 |
| HP_ARTHROPATHY | 30 | -0.66 | 0.97 | 1.00 |
| HP_JUVENILE_RHEUMATOID_ARTHRITIS | 11 | -0.58 | 0.94 | 1.00 |

**Fig 6. No compelling arthritic signatures after SCV vaccination.** (A) MSigDB contains 16 gene sets associated with arthritis/arthralgia. These gene sets were used in GSEA analyses against the complete pre-ranked gene list for MF vs SCVd7F (SP2 Table). (B) MF vs SCVd7F provided 22 DEGs (with 2 filters applied q<0.05 and count sum >6) of which 8 were up-regulated (Table S2Q). Three of these were also up-regulated DEGs for CHIKV arthritis day 7 post infection (Table S2O; q<0.05). Red text–gene products associated with pro-inflammation activities. Green text–gene products with anti-inflammatory activities. (C) The 22 DEGs analyzed by IPA *Diseases and Functions* feature (direct and indirect) (S2Q Table) and compared with the same annotations identified by IPA analysis of DEGs for CHIKV arthritis (S2O Table).

across all 6 samples >6, resulting in only 22 DEGs, of which 8 were up-regulated (S2Q Table). Of the 8 up-regulated DEGs, Tnfaip6 and Crispld2 have anti-inflammatory activities [179,180] and Thbs4 and Daglb have pro-inflammatory activities [181,182], with Gbp3 and Spon2 associated with antiviral responses [183,184] and Mrgprf associated with the itch response [185]. These results argue there were minimal transcriptional changes in joints after SCV-ZIKA/CHIK vaccination.

IPA *Diseases and Functions* analysis of the 22 DEGs returned a significant "Inflammatory response" signature with a negative z-score, and cellular infiltrate terms with low z-scores (Table S2Q). Comparison of the 22 DEGs with those reported for collagen induced arthritis (CIA) (GSE13071) [186] also identified no obvious concordance (S2Q Table). With only 22 DEGs such analyses are somewhat underpowered, so the entire gene list (S2P Table) was interrogated for the presence of arthritic signatures using pre-ranked GSEAs and the 16 available genes sets associated with arthritis/arthralgia available from MSigDB. No significant arthritic signatures were identified, with all q values >0.99 (Fig 6A). Thus, even when using the entire RNA-Seq derived data set, no indication of arthritic signatures were evident.

To the best of our knowledge there is no evidence to suggest that CHIKV antigens migrate to joints and cause arthropathy. Instead, CHIKV arthritis [187], and viral arthritides generally [188], are associated with replication of virus in the joints, with SCV-ZIKA/CHIK not detected in feet (Fig 1A). Nevertheless, after the experience with an attenuated CHIKV vaccine (TSI-GSD-218) that caused arthralgia in 5 of 58 vaccine recipients [42], arthropathy caused by

CHIKV vaccines remains a standing concern for regulators [41]. We thus also compared the DEGs and pathways identified for SCVd7F, with those identified for CHIKV arthritis [116]. We re-analyzed the FASTQ files generated in the latter publication (deposited in NCBI Bio-Project PRJNA431476) using STAR aligner and the more recent mouse MM10 genome build (GRCm38 Gencode vM23). The complete gene list for day 7 feet (peak CHIKV arthritis) is provided in S2O Table. Three of the 22 DEGs (Daglb, Tgtp2 and Gbp3) were also present in the up-regulated DEGs for day 7 feet of CHIKV infected mice (S2O Table); however, the fold change of the latter two were substantially higher after CHIKV infection than after SCV vaccination (Fig 6B). The four pathways identified by IPA *Diseases and Functions* analysis of the 22 DEGs (S2Q Table) were also present for CHIKV arthritis, but the z-scores and p-values for these annotations were very much lower for SCVd7F (Fig 6C).

Overall these results argue that SCV-ZIKA/CHIK vaccination was not associated with a compelling arthritic signature, even though the injection sites (quadriceps muscles) were in the same legs as the feet that were used to generate the MF vs SCVd7F gene set.

## Discussion

We provide herein a detailed injection site vaccinology analysis of a recombinant SCV vaccine in mouse muscle, to provide insights into vaccine gene expression, and host adjuvant signatures and immune responses. Of all the reads mapping to the SCV-ZIKA/CHIK vaccine, ≈20% mapped to the recombinant immunogens. IHC illustrated immunogen protein expression in skeletal muscle cells, with these cells showing histological signs of necrosis [63,64] and bioinformatics analyses indicating the presence of necrosis and necroptosis [62]. Adjuvant signatures were driven by TLRs, cytoplasmic RNA and DNA sensors and the inflammasome, with neutrophils potentially also contributing [53]. By day 7 vaccine transcripts and neutrophils were largely absent, and inflammation was abating, with the presence of what appeared to be tissue repair-associated eosinophils [189–193]. Although a previous live-attenuated CHIKV vaccine was associated with some arthropathy [42], no compelling arthritic signature was evident after SCV vaccination.

There was a marked concordance between (i) the z-scores for specific TLR and cytoplasmic sensor signatures identified at 12 hours post-vaccination and (ii) the relative importance of these pathways during the full course of infection, as gleaned from infection of knock-out mice with replication-competent poxviruses (summarized in S2E Table). The early dominant pathways identified by this injection site vaccinology approach was remarkably consistent with pathways previously published to be important for protection and/or immunopathology over the course of poxvirus infections (S2E Table). The concordance suggests that the early innate signatures identified herein are not overly dependent on (i) the ability of the vector to produce viral progeny [19] or (ii) the recombinant immunogen inserts. The concordance also suggests that these signatures may often be shared amongst different poxviruses and poxvirus vectors.

The ability to identify T and B cell response signatures on day 7 post-vaccination using GSEAs and BTMs, suggests RNA-Seq analyses of the injection site may also provide insights into the ensuing systemic adaptive immune responses. Migration of primed T cells to the site of infection is well described [194–196], so T cell signatures might be expected. However, plasma cells are ordinarily thought not to undergo such migration, arguing that the plasma cell signature may be due to non-specific migration, perhaps consistent with the relatively lower significance in the GSEA (Fig 5B, q = 0.03).

The immunogenicity of SCV, and likely poxvirus systems in general, would appear to be underpinned by the ability of such vectors to stimulate a broad range of pathways that are known to be stimulated by adjuvants that are already licensed for use in humans or are being

tested in humans. For instance, multiple TLR signatures were identified at 12 hours post vaccination (Fig 2A). The TLR4 agonist monophosphoryl lipid A is a component of Fendrix (hepatitis B vaccine) and Cervarix (human papilloma virus vaccine) (which are formulated in ASO4 adjuvant), as well as Shingrix (herpes zoster vaccine) (formulated in AS01 adjuvant). The dsRNA TLR3 agonists, Ampligen (Rintatolimod) and Hiltonol, also have well described adjuvant properties [197], with Hiltonol being tested in therapeutic cancer vaccine trials (ClinicalTrials.gov Identifier: NCT04345705 and NCT02423863). The TLR9 agonist, CpG oligonucleotide (CpG 1018), was recently approved as an adjuvant in Heplisav-B (hepatitis B vaccine). A range of cytoplasmic sensor signatures with known adjuvant activity were also identified (Fig 2B), including multiple inflammasome signatures; the best known human adjuvant, alum, is believed to mediate its activity via activation of the inflammasome [198]. Stimulation of cytoplasmic dsRNA sensors represents a key adjuvant activity for replicon-based RNA vaccines [199,200] and the utility of STING-activating adjuvants is being actively explored [201,202]. Although SCV does not generate viral progeny, it does replicate its DNA [19], perhaps explaining the dominant STING signature (Fig 2B). Virulent poxviruses inhibit STING activation via unknown factors, an inhibitory activity not found for MVA [203]. This activity is perhaps similarly absent for SCV or is inactive in mice.

A desirable feature for any vaccine is the avoidance of reactogenicity, a term describing a series of post-vaccination adverse events often associated with excessive injection site inflammation and systemic reactions such as fever [166,204]. A potential goal of transcriptome-based vaccinology is the identification of reactogenic signatures; however, consensus regarding the composition of such signatures, and/or when and where best to sample to obtain such signatures, has yet to be established [27]. CCL2 and CXCL10 up-regulation in peripheral blood was identified as potential biomarkers of vaccine-elicited adverse inflammation in mice after a number of different vaccines given i.m. [28]. These chemokines featured prominently at the injection site 12 hours after SCV vaccination (Fig 4B), although fold-change had reduced substantially by day 7 ($\log_2$ 4.12 to -0.73, and 8.35 to 3.17, respectively) (S2A Table and S2G Table). CCL2 is also induced by MVA [154], can be important for avoiding immunopathology [51], and is induced by the licensed adjuvants, Alum and MF59 [205]. CXCL10 is also induced by MVA [206] and by the licensed adjuvant, MF59 [207]. Given the extensive clinical safety record of MVA vaccination [208] and the lack of overt injection site reactogenicity or fever observed in NHPs after SCV-ZIKA/CHIK vaccination [22], CCL2 and CXCL10 up-regulation at the injection site would thus not appear to be compelling biomarkers for adverse events after MVA or SCV vaccination. Similarly, a whole-blood systemic adverse event signature for yellow fever 17D vaccination has been reported, with 32 up-regulated genes on day 1 (but not day 3) associated with a range of systemic adverse events (either within 24 hours or a median time post vaccination of 6 days) [209]. GSEAs illustrate that this signature is highly significantly present in the SCV12hQ gene list, and is also significantly present in the SCVd7Q gene list (S2R Table). All the core enriched genes (S2R Table) were type I IFN stimulated genes (by Interferome), with type I IFN stimulated gene induction clearly present at the SCV injection site (Fig 4A). However, MVA vaccination is also associated with short term (i) increases in local type I IFN responses [68,111] and (ii) elevated serum IFNα levels [210,211]. Of note, SCV encodes B19R/B18R, an inhibitor of type I IFN responses [148]. Unlike Yellow fever 17D, MVA and SCV vaccinations are not associated with significant viremias or viral dissemination, reducing the probability of excessive serum type I IFN responses and systemic adverse events; although pyrexia, headache, myalgia, nausea, fatigue and/or chills are seen in a small percentage of MVA vaccine recipients [166,208]. Clearly, adverse event signatures identified in peripheral blood, may not be overly informative for understanding adverse events at the injection site. In addition, not only the presence of specific gene transcripts but also the magnitude

of gene induction are likely to be important, with the latter not fully taken into account by GSEAs. In humans, sampling injection sites is difficult, although emerging micro-sampling techniques may provide new avenues [212]. Ultimately RNA-Seq of injection sites in animal models should be able to provide early warnings in the vaccine development process of potential reactogenicity issues. Herein we show that, although SCV retains the ability to replicate its DNA ([19], the injection site reactogenicity (like IMVANEX) [166] has largely resolved by day 7 post vaccination.

How might the information provided herein find utility for poxvirus vaccine design? The ZIKA and CHIK immunogens are inserted into B7R/B8R and A39R, respectively [18]. The 12 vector genes that were not expressed *in vivo* post-vaccination (S1B Table) offer other potential insertion sites for recombinant immunogens that would ostensibly have minimal impact on vaccine behavior. However, expression of these genes in human muscle might be checked, perhaps via use of human skeletal muscle organoids [213]. The multiple adjuvant pathway stimulated by SCV (Fig 2) might argue for a certain level of redundancy [211], which might allow certain inhibitors to be reintroduced with the aim of reducing reactogenicity, without compromising immunogenicity. For instance, B13R (also known as SPI-2) is absent in the Copenhagen strain of VACV and in SCV, and inhibits caspase I, a key protease for generation of bioactive IL1-β [214]. Reintroducing VACV IL-1 decoy receptors, SPI-2 (B13R) or B16R [215], into the vector may affect CD8 T cell responses [216,217], but have minimal effects on antibody responses [218]. Such secreted proteins should reduce the bioavailability of IL-1, a potent pyrogen, and may thereby reduce the risk of adverse events such as fever [144,166]. Recent sequencing of ancient Variola viruses from Viking corpses perhaps supports such strategies, as active expression of immune modulating genes may be associated with reduced pathogenicity [219,220]. Co-formulation of SCV with adjuvants [221,222] or encoding genetic adjuvants within SCV [223] might appear superfluous, given the large number of adjuvant pathways already being activated. Introducing apoptosis inhibitors to improve immunogenicity [14] may have minimal impact for i.m. injections of SCV (or other pox vectors) as skeletal muscle does not appear readily to undergo apoptosis [60,61], with B13R also well expressed at the injection site (S1B Table). The absence of the chemokine inhibitors B29R (C23L) (not expressed) and CCI (not present/functional) may contribute *inter alia* to effective recruitment of dendritic cells. Deletion of A41L might be tested to determine whether this would increase immunogenicity, given CCL21 recruits T cells and enhances T-cell responses [224]. Complement control proteins VCP (encoded by B27R) and C3 (encoded by C3L) were expressed at the mRNA level, with VCP deletion from VACV increasing anti-VACV antibody responses [13], suggesting deletion of these genes might enhance immunogenicity. One might consider deletion of N2L (an inhibitor of IRF3) as this was shown to improve the immunogenicity of a recombinant MVA vaccine [225] and also reduced the virulence of VACV [226]. However, the IRF3 signature is already very dominant (Fig 3B), so additional IRF3 activation (if possible) may not translate to significant improvements in immunogenicity. Deletion of C6L increased the immunogenicity of a recombinant MVA vaccine [17], presumably by relieving IRF3, IRF7 [105,106] and/or STAT2 inhibition [160]. However, C6L deletion could risk excessive type I IFN responses and increased reactogenicity, with excessive type I IFN responses associated with adverse events after administration of the yellow fever vaccine [209].

The presence of eosinophils in the resolving lesion on day 7 was unexpected. Eosinophils have been reported in pruritic papulovesicular eruptions in a case of generalized vaccinia after smallpox vaccination [227] and are well described as drivers of allergic diseases such as eosinophilic asthma [228,229]. However, recently a role for eosinophils in tissue repair and wound healing has emerged [189,190], particularly for muscle tissues [191–193]. That the eosinophils in these resolving post-vaccination lesions are distinct from inflammatory eosinophils is

supported by the absence of IL-5 mRNA expression (S2G Table), with anti-IL-5 therapy used for eosinophilic asthma [230]. Transcripts for eosinophil cationic protein (Ear1) and eosinophil peroxidase (Epx) (granule components of inflammatory eosinophils) were also not detected. Eotaxins (CCL11, CCL24 and CCL26) were not up-regulated, with IL4, IL13, IL3 and GM-CSF (CSF2) transcripts absent (S2G Table). Using GSEAs we were unable to find any eosinophil gene signatures in the full MQ vs SCVd7Q gene list, suggesting that signatures for inflammatory eosinophils are distinct from tissue repair-associated eosinophils, with the signatures for the latter yet to be defined.

A limitation of using mice to analyze VACV-based vaccines is that several VACV-encoded inhibitors are not active in mice (encoded by M2L, A38L) and the activity of others in mice is not known (D9R, H1L, C10L). The activity in mice of B28R and B29R is also not known, but these inhibitors are not active in the Copenhagen strain of VACV. Others were found not to be expressed in mice (C16L, C23L, C22L), although it is unclear whether they are poorly expressed by SCV, poorly expressed in muscle or poorly expressed in mice. How critical these genes are to the overall interpretations presented herein is difficult to assess, given the presence of multiple overlapping and potentially cross-compensating pathways. Another limitation of this study is that we have not established which signatures are associated with the SCV vector and those that are associated with the ZIKA and CHIK immunogens. The concordance with existing poxvirus literature (S2E Table) might argue that the dominant signatures are associated with the vector. This contention is supported by the general observation that recombinant proteins are poorly immunogenic and require adjuvant before they can induce significant immune responses [231,232]. CHIKV and ZIKV immunogens may assemble into virus-like-particles (VLPs) [233,234], but even VLPs often require formulation with adjuvants [235,236]. Should SCV be developed as a smallpox vaccine, separate injection site vaccinology studies might be warranted for SCV, which comprises the Copenhagen strain of VACV with D13L deleted [19]. Some changes in signatures from those reported herein might be expected as SCV would retain (i) A39R which encodes a protein with proinflammatory properties [237] and (ii) B7R/B8R which encode a virulence factor [238] and a chemokine receptor that regulates leukocyte trafficking [158]. SCV's DNA replication and protein expression may also be altered as VLP formation may *inter alia* compete for lipids with SCV [239–241], influence SCV-induced cell death [242] and/or modulate SCV's cytoskeletal rearrangements [243,244]. Also absent for SCV vaccination would be any competition for transcription-associated factors [245] imposed by the two synthetic strong early late promoters (Fig 1A).

An extensive history of poxvirus vector development has led to the first licensed recombinant poxvirus based vaccine for human use (MVA-BN-Filo) [12], with several others in late stage clinical trials. Injection site vaccinology may facilitate rationale refinement of pox vector design and contribute to progressing more such technologies towards registration and licensure.

## Methods

### Ethics statement

All mouse work was conducted in accordance with the "Australian code for the care and use of animals for scientific purposes" as defined by the National Health and Medical Research Council of Australia. Mouse work was approved by the QIMR Berghofer Medical Research Institute animal ethics committee (P2235 A1606-618M). Mice were euthanized using $CO_2$.

### The SCV-ZIKA/CHIK vaccine

SCV-ZIKA/CHIK was constructed as described [1] and manufactured as described [22]. Briefly, the CHIKV (Genbank: AM258992) and ZIKV (Genbank KU321639) structural

polyprotein cassettes were inserted into A39R and B8R/B7R gene loci, respectively. The vaccine was produced in a non-GMP SCS line (comprising CHO-S cells expressing D13L and CP77 [19]) using protein-free cell culture conditions. SCV-ZIKA/CHIK was released from infected cells by multiple freeze thaw cycles, cell debris removed by centrifugation, followed by sucrose cushion purification and resuspension in 10 mM Tris HCl pH 8, 150 mM NaCl and storage at -80˚C. The vaccine was tested for sterility (turbidity in broth medium) and for mycoplasma (by PCR).

## Mice and vaccination

Female C57BL/6J mice (6–8 weeks) were purchased from Animal Resources Center (Canning Vale, WA, Australia). Mice were anaesthetized with Isothesia NXT (Henry Schein Inc., Melville, NY, USA) and subsequently vaccinated once with 50 µl of 0.5 x $10^6$ pfu SCV-ZIKA/CHIK i.m. or Mock vaccinated (with PBS) into both quadriceps muscles as described [18].

## RNA-Seq

At the indicated times post vaccination, mice were euthanized using $CO_2$ and quadriceps muscles or feet placed individually into RNAlater (Life Technologies) overnight at 4˚C and then homogenized in TRIzol (Invitrogen) using 4 x 2.8 mm ceramic beads (MO BIO Inc., Carlsbad, USA) and a Precellys24 Tissue Homogeniser (Bertin Technologies, Montigny-le-Bretonneux, France) (6000 rpm on ice, 3 times 12 sec for feet and 2 times for 10 seconds for muscle). Homogenates were centrifuged (14,000 g x 15 min) and RNA extracted from the supernatants as per manufacturer's instructions. Following DNase treatment (RNAseq-Free DNAse Set (Qiagen)) and RNA purification (RNeasy MinElute Kit), RNA concentration and purity was determined by Nanodrop ND 1000 (NanoDrop Technologies Inc.). RNA samples were pooled so that for each group of 6 mice, 12 sets of quadriceps muscles or feet (severed at the bottom of the tibias after euthanasia) were used to create 3 biological replicates which contained equal amounts of RNA from 4 different mice. All replicates were then sent to the Australian Genome Research Facility (AGRF, Melbourne, Australia) for library preparation and sequencing. RNA integrity was assessed using the Bioanalyzer RNA 6000 Nano assay (Agilent) and libraries were prepared from 200 ng of total RNA using TruSeq Stranded mRNA library preparation kit (Illumina). The resulting libraries were assessed by TapeStation D1K TapeScreen assay (Agilent) and quantified by qPCR using the KAPA library quantification kit (Roche). Libraries were normalized to 2 nM and pooled for clustering on an Illumina cBot system using HiSeq PE Cluster Kit v4 reagents followed by sequencing on an Illumina HiSeq 2500 system with HiSeq SBS Kit v4 reagents with 100 bp paired-end reads.

## Mouse genome alignments and differential gene expression

Mapping to the mouse genome and differential expression analysis was conducted at AGRF under commercial contract using their in-house pipeline. The quality of the raw sequencing reads were assessed using FastQC and MultiQC. Adapters were trimmed using the TrimGalore (0.4.4) program and reads with a length <30 bp or quality <10 were removed. Filtered reads were aligned to the *Mus musculus* reference genome (mm10; GTF file GRCm38.6 – annotation release 105) using the STAR aligner (v2.5.3a) with default parameters plus a parameter to restrict multi-mapping reads ('—outFilterMultimapNmax 2'). Counts per gene were summarized using the featureCounts (v1.4.6-p5) utility in Subread. A counts matrix was generated from the collective samples using in-house scripts and input to R (3.5.0) for differential expression analysis. Differential expression analysis was undertaken using EdgeR (3.22.3) with default settings and no filters, given the importance of key genes with low transcript

abundance [116] and the small percentage of cells infected by SCV-ZIKA/CHIK in the quadriceps muscles (with whole quadriceps muscles harvested for RNA-Seq). Counts were converted to relative counts (CPM) and normalized using the TMM method and modelled using the likelihood ratio test, glmLRT().

### Ingenuity Pathway Analysis (IPA)

Up-Stream Regulator (USR), and *Diseases and Functions* features of Ingenuity Pathway Analysis (IPA) (QIAGEN) were used to interrogate the DEG lists using 'direct and indirect interaction' or 'direct only' interaction options.

### Read alignments to the mouse and viral genomes

Sequencing reads were assessed using FastQC [246] (v0.11.8) and trimmed using Cutadapt [247] (v2.3) to remove adapter sequences and low-quality bases. Trimmed reads were aligned using STAR [248] (v2.7.1a) to a combined reference that included the GRCm38 primary assembly and the GENCODE M23 gene model [249], VACV Copenhagen (M35027.1; 191737 bp), ZIKV (KU321639.1; 10676 bp), and CHIKV (AM258992.1; 11601 bp). Quality control metrics were computed using RNA-SeQC [250] (v1.1.8) and RSeQC [251] (v3.0.0). SAMtools [252] (v1.9) was used to obtain alignments to the coding sequences of mature peptide features of VACV, CHIKV and ZIKV.

### Kraken metagenomic sequence classification

RNA-Seq reads that were not assigned/mapped to any of the GRCm38, VACV, ZIKV or CHIKV reference genomes were analyzed using metagenomic sequence classification. Unmapped or unassigned reads were converted to fastq format using Bedtools version 2.26.0 [253]. Triplicates were concatenated to produce one read file per treatment. Exact-match database queries were performed using Kraken version 2.1.1 with the Minikraken2 version 2 database [254]. Data was visualized using Krona version 2.7.1 [255].

### Cytoscape and GSEAs using BTMs

Protein interaction networks of differentially expressed gene lists were visualized in Cytoscape (v3.7.2) [256]. Enrichment for biological processes, molecular functions, KEGG pathways and other gene ontology categories in DEG lists was elucidated using the STRING database [257].

Gene Set Enrichment Analysis (GSEA) [258] was performed on a desktop application (GSEA v4.0.3) and the GenePattern Public server [259] using the "GSEAPreranked" module. Gene sets for these analyses were obtained from blood transcription modules (BTM_for_G-SEA_20131008.gmt, n = 346) described previously [165]. BTM gene sets were used to run GSEAs on the pre-ranked (by fold change) gene list for MQ vs SCV12hQ (S2A Table) and MQ vs SCVd7Q (S2G Table).

### GSEAs using MSigDB gene sets

The complete Molecular Signatures Database (MSigDB) v7.2 gene set collection (31,120 gene sets) (msigdb.v7.2.symbols.gmt: https://www.gsea-msigdb.org/gsea/msigdb/download_file.jsp?filePath=/msigdb/release/7.2/msigdb.v7.2.symbols.gmt) was used to run GSEAs on pre-ranked gene lists for MQ vs SCV12hQ (S2A Table) and MQ vs SCVd7Q (S2G Table).

## Histology and immunohistochemistry

H&E staining was undertaken as described previously [52]. IHC for neutrophils was undertaken as described [52] using Ly6G primary antibody (Abcam Anti-Mouse Neutrophil antibody Clone: NIMP-R14 cat. No. ab2557, Cambridge, UK) and Ly6 secondary antibody (Biocare Medical Rat on Mouse HRP Polymer cat. no. RT517L, Concord, CA USA). ApopTag staining used the Millipore ApopTag Peroxidase In Situ Apoptosis Detection kit (cat. No. S7100 Temecula, CA, USA). IHC for CHIKV capsid (monoclonal antibody 5.5G9 [54]) and ZIKA envelope (monoclonal antibody 4G2 [55], was undertaken as described using NovaRed secondary antibody (Vector Laboratories ImmPACT NovaRed Peroxidase Substrate Kit cat. No. SK-4805 Burlingame, CA, USA). Slides were digitally scanned using Aperio AT Turbo (Leica Biosystems).

## Statistics

Statistical analysis of experimental data was performed using IBM SPSS Statistics for Windows, Version 19.0 (IBM Corp., Armonk, New York, USA). The paired t-test was used when the difference in variances was <4, skewness was >2 and kurtosis was <2. Otherwise the non-parametric Wilcoxon Signed Rank tests was used.

## Supporting information

**S1 Fig. RNA-Seq.** (A) Time line of experiment. (B) Pooling strategy for replicates. (C) Reads and percent of reads assigned to the mouse genome. (D) Boxplot of Log counts (normalized). Boxplots shows similar distributions of read counts amongst samples within and between groups. Boxes are 1st & 3rd quartile; whiskers range (no outliers). (E) MDS plot showing (i) clear separation between feet and quadriceps muscle groups, (ii) tight clustering of triplicates for MQ, SCVd7Q and SCV12hQ groups, (iii) clear separation between MQ, SCVd7Q and SCV12hQ groups, and (iv) poor separation between MF and SCVd7F (consistent with the low number of DEGs). (F) Smear plots of the differentially expressed genes for the three comparisons. Red–FDR <0.05. Blue lines represent fold change of 2.
(TIF)

**S2 Fig. Bowtie2 alignments to viral genomes.** Raw FASTQ files were assessed for quality using FastQC and MultiQC tools. Sequencing adapters were trimmed using Trimmomatic (0.36.6) [1] where reads with an average quality score over a 4 base sliding window of less than 20 were removed. Trimmed reads were aligned using Bowtie2 (v2.3.4.1)[2] to a combined reference that included the GRCm38 primary assembly and the GENCODE M23 gene model, Vaccinia virus Copenhagen (M35027.1), Zika virus strain Zika SPH2015 (KU321639.1), and chikungunya virus (AM258992.1). Primary proper pair reads aligned to viral features, including CDS and mature peptide features, were counted using SAMtools (v1.9). (A) Bar graph of vaccine read counts expressed as a percentage of reads aligning to the mouse genome. (B). Raw data for A. [1]Bolger AM, Lohse M, Usadel B. Trimmomatic: a flexible trimmer for Illumina sequence data. Bioinformatics. 2014;30(15):2114–20. [2]Langmead B, Salzberg SL. Fast gapped-read alignment with Bowtie 2. Nature Methods. 2012;9(4):357–9.
(TIF)

**S3 Fig. High resolution H&E and IHC control.** (A) High resolution image of Fig 1D showing the striations in health muscle cells (S) and above these, paler muscle cells that have lost their striated appearance. Small condensed pyknotic nuclei are indicated by arrows. (B) Expanded view of IHC staining shown in Fig 1F, with positive staining indicated by dotted oval. (C)

Staining of a parallel section to that shown in B stained with a control antibody.
(TIFF)

**S4 Fig. A549 cells infected with SCV-ZIKA/CHIK.** 48–72 hours after SCV-ZIKA/CHIK infection of A549 cells *in vitro*, morphological features characteristic of apoptosis (condensation of chromatin) were clearly evident (top row) after staining with Hoechst 33342 [1]. Bottom image shows uninfected controls. [1]Linn *et al*. Complete removal of mycoplasma from viral preparations using solvent extraction. J Virol Methods. 1995. 52(1–2):51–4.
(TIF)

**S5 Fig. GSEAs with MSigDB gene sets.** All genes for MQ vs SCV12hQ and MQ vs SCVd7Q were pre-ranked and GSEAs run for the >31,000 genes set available in MSigDB. Listed are vaccine and virus infection signatures where q<0.05.
(TIF)

**S6 Fig. Unmapped/unassigned reads analysed by Kraken.** Reads that were not assigned by the STAR aligner to mouse or vaccine genomes for MQ, SCV12hQ and SCVd7Q were analysed by Kraken, a metagenomic sequence classification tool. The output for all classified reads (All) was dominated by human sequences (*Homo sapiens*), *Pasteurella multocida* (a commensal of dogs, cats and rabbits), and murine retroviruses (primarily *Mus musculus* mobilized endogenous polytropic provirus). Mice in our animal house facility routinely test negative for *Pasteurella* species, so this contamination is unlikely to have originated from the mice. About 10% of the mouse genome is made up of endogenous retroviruses, with multi-mapped reads (reads that align to multiple locations in the mouse genome) left unassigned by the STAR aligner.
(TIF)

**S1 Table. Alignment of reads to the SCV/ZIKA/CHIK vaccine genome.**
(XLSX)

**S2 Table. Gene lists and bioinformatics of mouse responses.**
(XLSX)

## Acknowledgments

We would like to thank the animal house staff and Histology and Imaging Services at QIMR Berghofer for their assistance. We also thank Dr T Larcher (Institut National de Recherche Agronomique, France) for help interpreting histopathology.

## Author Contributions

**Conceptualization:** Andreas Suhrbier.

**Data curation:** Andreas Suhrbier.

**Formal analysis:** Jessamine E. Hazlewood, Troy Dumenil, Andrii Slonchak, Stephen H. Kazakoff, Lesley-Ann Gray, Paul M. Howley, Kexin Yan.

**Funding acquisition:** Andreas Suhrbier.

**Investigation:** Jessamine E. Hazlewood, Troy Dumenil, Thuy T. Le, Stephen H. Kazakoff, Lesley-Ann Gray, Natalie A. Prow.

**Methodology:** Andrii Slonchak, Stephen H. Kazakoff, Ann-Marie Patch, Lesley-Ann Gray, Paul M. Howley, Andreas Suhrbier.

**Project administration:** Andreas Suhrbier.

**Resources:** Andrii Slonchak, Stephen H. Kazakoff, Ann-Marie Patch, Lesley-Ann Gray, Paul M. Howley, Liang Liu, John D. Hayball, Natalie A. Prow.

**Software:** Andrii Slonchak, Stephen H. Kazakoff, Ann-Marie Patch.

**Supervision:** Andreas Suhrbier.

**Validation:** Andreas Suhrbier.

**Visualization:** Andreas Suhrbier.

**Writing – original draft:** Andreas Suhrbier.

**Writing – review & editing:** Jessamine E. Hazlewood, Troy Dumenil, Ann-Marie Patch, Daniel J. Rawle, Andreas Suhrbier.

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
