## [Decision Letter · Decision Letter 0]

12 Nov 2020

Dear Prof Suhrbier,

Thank you very much for submitting your manuscript "Systems vaccinology analysis of a recombinant vaccinia-based vector reveals diverse innate immune signatures at the injection site" for consideration at PLOS Pathogens. As with all papers reviewed by the journal, your manuscript was reviewed by members of the editorial board and by several independent reviewers. In light of the reviews (below this email), we would like to invite the resubmission of a significantly-revised version that takes into account the reviewers' comments.

We cannot make any decision about publication until we have seen the revised manuscript and your response to the reviewers' comments. Your revised manuscript is also likely to be sent to reviewers for further evaluation.

Sincerely,

Grant McFadden, PhD

Guest Editor

PLOS Pathogens

Klaus Früh

Section Editor

PLOS Pathogens

Kasturi Haldar

Editor-in-Chief

PLOS Pathogens

orcid.org/0000-0001-5065-158X

Michael Malim

Editor-in-Chief

PLOS Pathogens

orcid.org/0000-0002-7699-2064

Reviewer's Responses to Questions

**Part I - Summary**

Reviewer #1: In this manuscript the authors have studied the changes in gene expression after vaccination in mice with a multiplication defective poxvirus-based vector system called Sementis Copenhagen vector. This vector expresses the structural genes of both CHIKV and ZIKV as immunogen. The vaccination was done by intramuscular injection (IM) and muscles were harvested at 12 hours and 7 days post vaccination. The authors performed RNA-Seq and bioinformatics to characterize the injection site innate responses and expression of the recombinant immunogens.

This reviewer thinks there are several issues with the study design:

1. The authors should add vector only control, to differentiate the changes that are due to the expression of immunogens. It is known that the vector itself will activate or suppress many of the innate response markers.

2. Does analysis of muscle only (injection site) represent systems vaccinology? What about detection of immune responses in other key organs? B cells, T cells? It is not surprising that at the injection site after 7 days almost nothing is detected.

3. What was the rationale of selecting only these two time points?

4. Validation of data?

Reviewer #2: The authors of this manuscript used RNAseq to analyze changes in the abundance of gene transcripts in mice vaccinated (IM) with a recombinant VAC strain encoding Chikungunya and Zika virus antigens. Two time points were evaluated, 12 hr and 7 days post-vaccination. Although the conclusions were perhaps not too surprising, the investigation and subsequent analyses provided a comprehensive snapshot of the innate and adaptive response to such an infection.

I found this an interesting manuscript, excellent science, and intriguing observations. The results would be of interest to anyone trying to understand how a recombinant poxvirus might vector an antigen. Lots to chew on.

My criticisms are not so much scientific as concerning the manuscript itself. As a general observation it often reads like a review of vaccinia virus gene functions, which has value, but at times isn't too relevant to the interpretation of these data.

In particular there are many references to how VACV protein "X" inhibits protein "Y", but then it is often unclear why that should have any impact on the level of the relevant cellular transcript(s). For example, on lines 272-3 it's said that "The DNA sensor PRKDC/DNA-PK has a z-score of zero, perhaps due to the inhibitory activity of C4 (encoded by C4L)...". Binding wouldn't necessarily affect transcript levels. Unless it's known that there is a feedback loop wherein the stated virus-host interaction inhibits expression of its binding target, such an interaction isn't really of relevance to the paper. In a related discussion (lines 316-9), the authors note that K1 and A49 block NFkB signaling in the context of the fact that IPA USR analysis detected a "dominant NF-kB signature". Doesn't such a signature imply that such promoter elements are active? This seems to be a non-sequitur, unless one is willing to hypothesize that K1 and A49 aren't actually working as we suppose.

I would ask that the authors take a close look at wherever they have described a VACV gene function, and ask themselves whether the citations (over 200 of them) help the reader gain a better understanding of the data reported in this manuscript.

Some minor observations:

line 72-3. Should ACAM2000 be included? It's arguably more relevant than racoonpox.

line 123. Are there bacterial reads in the remaining 8.4%? That is, is there any signature associated with secondary microbial infection?

line 154. Should probably read "gene" not "protein"

line 177. Unless I missed it, somewhere it should be stated explicitly that all the reported effects were absent in mock vaccinated animals.

line 249-50. The sentence is so full of abbreviations as to render it incomprehensible. There are others like it. Are all of these abbreviations really needed?

lines 526-30. I wasn't quite sure what reason exists to review different adjuvants. Please clarify.

Fig 5I. Are some of the muscle signatures related to the damage caused by the injection? Is there any data that would allow a comparison of the mock vaccinated (MQ) versus untreated tissue?

Reviewer #3: Review on Hazlewood et al., PPATHOGENS-D-20-01887

This manuscript reports from the preclinical characterization of a recombinant vaccinia virus candidate vaccine (Sementis Copenhagen Vector, SCV) co-expressing structural antigens of chikungunya virus (CHIKV) and Zika virus (ZIKV) and referred to as SCV-ZIKA/CHIK. Groups of C57BL/6 mice were intramuscularly vaccinated with 10E6 plaque-forming-units (pfu) SCV-ZIKA/CHIK or saline (PBS). At two time points post vaccination (12 hours or 7 days) organ samples (quadriceps muscles, feet) were taken and served to prepare total RNA for next-generation sequencing (NGS). The sequencing reads were aligned to mouse and viral genomes and used in a systems vaccinology approach. Aim was to assess host responses and viral gene expression at the site of vaccine inoculation. The paper provides a detailed description of the various gene expression profiles with clearly the most distinct data revealed at 12h post vaccination. With regard to viral (SCV-ZIKA/CHIK-specific) gene expression an interesting and even surprising finding was that about 20% of all transcripts mapped to the recombinant target gene sequences. The detected host responses demonstrate the activated expression of various genes associated with pathogen recognition and the activation of innate immunity (e.g. signatures for PRRs, their signaling molecules, cytokine, chemokines). Together with the data from the histopathological investigation the study provides a solid data set describing the inoculation site impact of the SCV-ZIKA/CHIK infection. The experimental work and data analyses are extensive and appear mostly well done. The many observations are extensively discussed and appear sometimes somewhat speculative. Here, it is regrettable that the data analysis just compares the inoculations with SCV-ZIKA/CHIK or saline. From the perspective of poxvirus biology and poxvirus vector vaccine development the study would have been even more informative when including the analysis of inoculations with the original Copenhagen strain of VACV and with an empty SCV. The following specific points should be considered:

**Part II – Major Issues: Key Experiments Required for Acceptance**

Reviewer #1: This reviewer thinks there are several issues with the study design:

1. The authors should add vector only control, to differentiate the changes that are due to the expression of immunogens. It is known that the vector itself will activate or suppress many of the innate response markers.

2. Does analysis of muscle only (injection site) represent systems vaccinology? What about detection of immune responses in other key organs? B cells, T cells? It is not surprising that at the injection site after 7 days almost nothing is detected.

3. What was the rationale of selecting only these two time points?

4. Validation of data?

Reviewer #2: N/A

Reviewer #3: 1. Results (p7, ln 153). The promising finding of abundant recombinant immunogen transcripts (~20%) among the total viral transcripts would deserve further attention. It would be interesting to determine if the particular replication-defective life cycle of SCV contributes to this expression phenotype.

2. Results, histopathology (Fig. 1D). The lesions described here appear quite extensive seen the low dose of virus inoculated into the muscle. Larger magnifications of the photograph should be provided to allow for a clear histopathological picture/analysis of the potentially damaged tissue.

3. Results, immunohistochemistry (Fig. 1F). This type of analysis is always prone to background staining and requires appropriate negative controls. The authors should demonstrate that there is no background staining in muscle tissues infected with empty SCV.

4. Results (p20-21), presence of eosinophils and absence of arthritic signature (Fig. 5). Is the assumption that the delivery of CHIKV antigens might result in a potentially harmful inflammatory response or in an autoimmune-like reaction (at day 7 post vac)? The rational to ask these questions needs a clear explanation (other than reference to CHIKV infection in general). Is there any evidence of adverse events caused by these subunit antigens? If so, the comparative analysis of empty SCV inoculations seems mandatory to address these questions.

5. Discussion (p22, ln 513 ff). The statement that the observed early innate signatures may be shared amongst different poxviruses is interesting but speculative. Here, it would have been very interesting to directly compare data e.g. from inoculations with the VACV Copenhagen as the direct ancestor of SCV.

**Part III – Minor Issues: Editorial and Data Presentation Modifications**

Reviewer #1: (No Response)

Reviewer #2: Please see above.

Reviewer #3: 1. Author summary (p3, ln 58). Recently, a recombinant MVA vaccine has been licensed by the European Commission as part of Janssen’s two component Ebola vector vaccine using ZABDENO together with MVABEA.

2. Results (p7, ln 146). What is the rational to hypothesize that the SCV vaccine following intramuscular inoculation could disseminate and persist in joint tissues (feet)?

3. Discussion (p23, ln 544ff; p25, ln588ff). The identification of vaccine associated reactogenic (or protective) signatures is an often postulated, potentially attractive goal of systems vaccinology approaches. To contribute to this objective it seems still necessary generate data from appropriate comparative experimental settings. E.g. Mutant MVA with an inactivated IL1-ß receptor elicits improved protective immunity and enhanced CD8 T cell responses (Staib 2005, JGV 86:1997; Cottingham 2008 PLoSOne 3:e1638.). Thus, it could be informative to directly compare the signatures of such a mutant virus to conventional MVA.

4. Discussion (p25, ln576). MVA does replicate its DNA in (most) non-permissive cells.

5. Discussion. Reference 193 (p26, ln610) needs checking.

6. Discussion (p27, ln632). Recently, a recombinant MVA vaccine has been licensed by the European Commission as part of Janssen’s two component Ebola vector vaccine using ZABDENO together with MVABEA.

PLOS authors have the option to publish the peer review history of their article (what does this mean?). If published, this will include your full peer review and any attached files.

Reviewer #1: No

Reviewer #2: No

Reviewer #3: No
---

## [Editor Report · Decision Letter 1]

4 Dec 2020

Dear Prof Suhrbier,

We are pleased to inform you that your manuscript 'Injection site vaccinology of a recombinant vaccinia-based vector reveals diverse innate immune signatures' has been provisionally accepted for publication in PLOS Pathogens.

Best regards,

Grant McFadden, PhD

Guest Editor

PLOS Pathogens

Klaus Früh

Section Editor

PLOS Pathogens

Kasturi Haldar

Editor-in-Chief

PLOS Pathogens

orcid.org/0000-0001-5065-158X

Michael Malim

Editor-in-Chief

PLOS Pathogens

orcid.org/0000-0002-7699-2064
---

## [Editor Report · Acceptance letter]

7 Jan 2021

Dear Prof Suhrbier,

We are delighted to inform you that your manuscript, "Injection site vaccinology of a recombinant vaccinia-based vector reveals diverse innate immune signatures," has been formally accepted for publication in PLOS Pathogens.

Best regards,

Kasturi Haldar

Editor-in-Chief

PLOS Pathogens

orcid.org/0000-0001-5065-158X

Michael Malim

Editor-in-Chief

PLOS Pathogens

orcid.org/0000-0002-7699-2064